# Drug Target Identification and Drug Repurposing in Psoriasis through Systems Biology Approach, DNN-Based DTI Model and Genome-Wide Microarray Data

**DOI:** 10.3390/ijms241210033

**Published:** 2023-06-12

**Authors:** Yu-Ping Zhan, Bor-Sen Chen

**Affiliations:** Laboratory of Automatic Control, Signal Processing and Systems Biology, Department of Electrical Engineering, National Tsing Hua University, Hsinchu 30013, Taiwan

**Keywords:** pathogenic mechanism of psoriasis, systems biology method, core signaling pathways, DNN-based DTI model, drug design specifications, multi-molecule drug, bid data mining, DTI databases

## Abstract

Psoriasis is a chronic skin disease that affects millions of people worldwide. In 2014, psoriasis was recognized by the World Health Organization (WHO) as a serious non-communicable disease. In this study, a systems biology approach was used to investigate the underlying pathogenic mechanism of psoriasis and identify the potential drug targets for therapeutic treatment. The study involved the construction of a candidate genome-wide genetic and epigenetic network (GWGEN) through big data mining, followed by the identification of real GWGENs of psoriatic and non-psoriatic using system identification and system order detection methods. Core GWGENs were extracted from real GWGENs using the Principal Network Projection (PNP) method, and the corresponding core signaling pathways were annotated using the Kyoto Encyclopedia of Genes and Genomes (KEGG) pathways. Comparing core signaling pathways of psoriasis and non-psoriasis and their downstream cellular dysfunctions, STAT3, CEBPB, NF-κB, and FOXO1 are identified as significant biomarkers of pathogenic mechanism and considered as drug targets for the therapeutic treatment of psoriasis. Then, a deep neural network (DNN)-based drug-target interaction (DTI) model was trained by the DTI dataset to predict candidate molecular drugs. By considering adequate regulatory ability, toxicity, and sensitivity as drug design specifications, Naringin, Butein, and Betulinic acid were selected from the candidate molecular drugs and combined into potential multi-molecule drugs for the treatment of psoriasis.

## 1. Introduction

Psoriasis is a common chronic disease, which is an autoimmune disease. Its main symptoms are erythema, scales, dryness, and other lesions on the skin surface, usually affecting the scalp, elbows, knees, and back, and may be accompanied by pain and itching and other discomfort. Microscopically, psoriasis is characterized by markedly increased proliferation and incomplete differentiation of the epidermis, a marked increase in cutaneous blood flow, and leukocytic infiltration of the papillary dermis and the epidermis. In addition, psoriasis may also affect the nails and joints, causing symptoms such as deformed nails and joint pain. Psoriasis is a chronic disease, its symptoms may last for many years or lifelong, and it often presents a recurring state of incessant advances and retreats [1]. Psoriasis may also cause some psychologically related symptoms. Because the symptoms of psoriasis often appear on the skin, patients often feel distressed and insecure about their appearance, which in turn affects their mental health. Studies have shown that people with psoriasis may have higher levels of depression and anxiety and may even experience suicidal thoughts and behaviors [2]. Psoriasis is a global disease, and approximately 1% to 3% of the global population suffers from this disease, among which there are more patients in Europe and the United States and relatively few in Asia. It is estimated that approximately 125 million people worldwide currently suffer from psoriasis [3]. In 2014, World Health Organization (WHO) member states recognized psoriasis as a serious non-communicable disease and adopted resolution WHA 67.9, which called for multilateral efforts to raise awareness and combat stigma [4].

Psoriasis is an autoimmune disease, and its pathogenesis is not fully understood, but current research shows that a combination of factors may lead to the occurrence of psoriasis. First of all, genetic factors are one of the important causes of psoriasis. Research shows that family members with psoriasis are at higher risk than the general population [5]. There are specific gene variants that may be associated with the development of psoriasis, but no single gene mutation that is strongly associated with psoriasis has been found so far [6]. Secondly, environmental factors are also one of the causes of psoriasis. Environmental factors include stress, infection, smoking, alcohol abuse, overuse of certain drugs, overexposure to sunlight, etc. These factors may affect the function of the immune system, which can lead to the development of psoriasis [7]. In addition, psoriasis may also be related to abnormalities in the immune system. Under normal circumstances, the immune system attacks and destroys foreign bodies and abnormal cells to protect human health. However, in people with psoriasis, the immune system attacks the skin cells, causing them to overgrow and eventually form psoriatic plaques. In conclusion, psoriasis is an autoimmune disease, and its pathogenesis involves many factors, including genetic factors, environmental factors, and immune system abnormalities. Therefore, systematic genetic and epigenetic research, as well as genome-wide microarray data exploration and comparison of psoriasis and non-psoriasis, can help us better understand the etiology and treatment of psoriasis.

There is currently no complete cure for psoriasis, but there are a variety of medications that can help manage symptoms and improve the patient’s quality of life. The followings are common psoriasis medications: topical steroids suppress the immune system response and are widely used in the treatment of psoriasis. Topical steroids, including vitamin D analogs and topical corticosteroids, can reduce inflammation and control cell proliferation, reducing symptoms. However, long-term use of steroids increases the risk of side effects such as localized skin thinning and pigmentation [8]. Phototherapy is through ultraviolet B irradiation (UVB), narrow-wave ultraviolet B irradiation, and other methods. However, long-term side effects of UV exposure include pigmentary disorder, photoaging, cataracts, and carcinogenesis. Photodermatitis usually occurs in UV radiation as UV-induced DNA damage or mutation results in the activation of oncogenes or silencing of tumor-suppressor genes, which are closely related to the pathogenesis of skin squamous-cell carcinoma [9,10]. Immunosuppressants can suppress the immune system response and reduce inflammation and skin cell proliferation. These drugs are commonly used to treat severe psoriasis but can cause damage to the liver, kidneys, immune system, etc., and increase the risk of infection and disease [11]. Biological agents include TNF inhibitors, IL-23 inhibitors, etc. These medications are commonly used to treat severe psoriasis and can be effective in reducing symptoms and plaque size. However, biological agents affect the immune system to cause side effects, such as infection, fatigue, and approx. 30% of patients do not respond well to the therapy [12]. At present, there does not exist an effective medication to treat psoriasis. The approach is to combine systems medicine methods with some drug design specifications to improve the drug design discovery and design for the therapeutic treatment of psoriasis.

Drug discovery is a time- and resource-intensive process, often costing billions of dollars and over a decade [13]. Pharmaceutical companies spend significant resources conducting experiments to understand drug properties and their interactions with target molecules. Furthermore, drug efficacy and safety must also be considered, leading to extensive animal and clinical trials [14]. Despite these efforts, drug development failures are still common due to poor clinical outcomes [15]. Therefore, more efficient and systematic approaches to drug design are needed. Drug-target interaction (DTI) prediction is an important process in drug design and repurposing that can narrow down the list of drug candidates [16]. Traditional DTI prediction methods mainly include ligand-based methods and docking-based methods, which are used to predict interactions based on the similarity between proteins and ligands. In recent years, DTI prediction methods based on machine learning have developed rapidly [17,18,19,20]. Drug repurposing, which refers to the use of a drug for an indication other than its original approval, can yield a wealth of data for research and can also reduce the need for additional studies of pharmacokinetic properties and toxicity. A drug combination is a multi-molecule drug that can improve the efficacy of each molecular drug in the combination and reduce the toxicity, drug resistance, and side effects of patients [17]. DTI prediction for drug targets based on deep neural network (DNN) via the large DTI database can be applied to systemic drug design and discovery and to select adequate multi-molecule drugs for the treatment of various diseases from the perspective of drug repurposing and drug combination [18].

The goal of this study is to investigate core signaling pathways through the systems biology method, identify significant biomarkers for the pathogenic mechanism of psoriasis as drug targets, and employ a DNN-based DTI model to predict and design a multi-molecule drug for the treatment of psoriasis as shown in Figure 1. First, the genome-wide genetic and epigenetic networks (GWGENs) of psoriasis and non-psoriasis are constructed [19]. For the annotation of KEGG pathways, GWGENs of psoriasis and non-psoriasis are extracted by the principal network projection (PNP) method into two core GWGENs with 6000 significant nodes, as shown in Figure 2 and Figure 3, respectively. Then, two core GWGENs are annotated by KEGG pathways to core signaling pathways of psoriasis and non-psoriasis in Figure 4. Comparing core signaling pathways of psoriasis and non-psoriasis and their downstream cellular dysfunctions, the significant biomarkers for the pathogenic mechanism of psoriasis are identified as drug targets of therapeutic treatment. In order to improve the efficiency and success rate of drug discovery and design for the therapeutic treatment of psoriasis, a deep neural network (DNN)-based drug-target interaction (DTI) prediction model was proposed and trained by a large amount of DTI databases to predict candidate molecular drugs to target their significant biomarkers of psoriasis, as shown in Figure 5. The drug design specifications based on the regulation ability, toxicity, and sensitivity of drugs were employed for screening potential molecular drugs Naringin, Butein, and Betulinic acid, which can be combined as a multiple-molecule drug for the treatment of psoriasis.

The DNN-based DTI model method has significant advantages over traditional computational drug discovery methods [20]. It can learn the molecular features of drugs and targets and make predictions from a large number of drug and protein interactions [17]. As shown in the flowchart of multi-molecule drug design for psoriasis in Figure 5, the DNN-based DTI model will be trained using publicly available biological and chemical datasets to predict candidate drugs for biomarkers (drug targets) with accuracy and reliability. In addition to the development of single molecular drugs, drug repurposing and drug combinations are also the focus of our research. Drug repurposing can reduce the cost and time of drug development by harnessing the efficacy of existing molecular drugs to treat other diseases or conditions [17]. Molecular drug combinations can use more than one molecular drug to enhance the effect of treatment, thereby increasing the success rate of treatment and reducing side effects. Finally, our findings can provide a new perspective and approach for systems drug design and discovery, thereby opening up new possibilities for the systematic development and discovery of a multiple-molecular drug for therapeutic treatments of psoriasis and other diseases.

## 2. Results

### 2.1. An Overview of Systems Biology Approaches for the Study of Pathogenic Mechanisms and Systematic Drug Discovery and Design for the Treatment of Psoriasis

In this study, the goal is to investigate the pathogenic mechanism of psoriasis by systems biology methods in Section 4.3, Section 4.4 and Section 4.5 to identify significant biomarkers of pathogenesis as drugs targets of psoriasis and then use the drug-target interaction data to train DNN-based drug-target interaction (DTI) model to predict potential multi-molecule drug to target the significant biomarkers as shown in Figure 5. First, in order to realize the pathogenic mechanism to identify its significant biomarkers as drug targets for the therapeutic treatment of psoriasis, we constructed candidate GWGEN from the database (GSE117468) through the big data mining method. Candidate GWGEN is a binary matrix. The matrix will record 1 if there is an interaction between two proteins/genes and record 0 if two nodes have no interaction. The nodes of the candidate GWGEN are divided into several groups: proteins, receptors, transcription factors (TFs), genes, miRNAs, and lncRNAs. Then, candidate GWGENs are pruned to real GWGENs of psoriasis and non-psoriasis in Figure 2 using their corresponding microarray data by system identification in Equations (1)–(20) to trim off false positives in candidate GWGEN. Here, we utilize the Akaike information criteria (AIC) [21] in Equations (21)–(28) to perform the system order detection method to trim off the false-positive interactions and obtain real GWGENs of psoriasis and non-psoriasis. Although real GWGENs have been condensed, they are still too complex to be annotated by the KEGG pathways, which can only annotate 6000 molecules at most. Third, core GWGENs with 6000 nodes are extracted from real GWGENs. To investigate the significant pathways of psoriasis and non-psoriasis, we extract 6000 key nodes from real GWGENs and obtain core GWGENs in Figure 3 through the principal network projection (PNP) method in Equations (29)–(34). In the study, the top-ranked 6000 nodes of core GWGENs contain 85% network energy of real GWGENs. We plot the real and core GWGENs of psoriasis and non-psoriasis with the network visualization software Cytoscape in Figure 2 and Figure 3, respectively. The detail of nodes and edges in the candidate, real, and core GWGENs are given in Table 1. Fourth, based on the enrichment analysis of KEGG pathways in Table 2 and Table 3 for core GWGENs of psoriasis and non-psoriasis, respectively, we have established common and specific core signaling pathways for psoriasis and non-psoriasis and investigated the pathogenic mechanisms involved in the pathogenesis of psoriasis as shown in Figure 4. Important biomarkers of the pathogenesis of psoriasis were identified as drug targets based on the investigation of these core pathways and their targeted genes leading to downstream cellular dysfunctions of psoriasis. Fifth, for drug discovery and design, we proposed a DNN-based DTI model using the existing drug-target interaction (DTI) database as the training set in Equations (35)–(45) to predict candidate drugs for the drug targets of psoriasis as shown in Figure 5. Furthermore, we consider the regulation ability, sensitivity, and toxicity of candidate drugs as drug design specifications to select potential drugs to be combined as a multi-molecule drug for the therapeutic treatment of psoriasis. The flowchart in Figure 1 illustrates the process of establishing candidate, real, and core GWGENs, and then identifying core signaling pathways in psoriasis and non-psoriasis to investigate significant biomarkers of pathogenic mechanisms as drug targets and then predicting psoriasis. A more detailed description of the result will be given in the following subsections.

### 2.2. The Core Signaling Pathways and Their Downstream Cellular Dysfunctions to Investigate Molecular Pathogenesis of Psoriasis

In Figure 4, the common and specific core signaling pathways and their downstream cellular dysfunctions between psoriasis and non-psoriasis are given. Then, the specific core signaling pathways and their downstream cellular dysfunctions of psoriasis were investigated to identify biomarkers of the pathogenesis of psoriasis. The microenvironmental factor interleukin 6 (IL-6) is produced by keratinocytes and leukocytes. It is a cytokine that is a major mediator of the host response to tissue injury and infection. High expression of IL6 could directly cause psoriatic epidermal hyperplasia and affect the function of dermal inflammatory cells [22]. It plays a significant role in psoriasis by linking keratinocyte proliferation with immune activation and tissue inflammation. The receptor IL6R receives the microenvironment factor IL6 to activate JAK1 and STAT3 to upregulate TF STAT3 [23]. The Janus Kinase–Signal Transducer and Activator of Transcription (JAK–STAT) pathway plays a significant role in the intracellular signaling of cytokines of numerous cellular processes, which is important in both normal and pathological states of immune-mediated inflammatory diseases [24]. The transduction of STAT3 is involved in the maintenance of postnatal interactions between epithelial and mesenchymal compartments [25]. The upregulated TF STAT3 will overexpress target genes *VEGF*, *COX-2*, and *BCL-XL* [26]. The overexpression of VEGF induces angiogenesis in psoriasis [27,28]. *BCL-XL* is an anti-apoptotic gene. Overexpression of *BCL-XL* prevents apoptosis and thickens the epidermis [29,30]. *COX-2* has been reported to be involved in acute inflammation [31].

In Figure 4, another microenvironment factor is IGF1. The pathogenesis of psoriasis is due to the activation of immune cells and their secretion of cytokines, chemokines, growth factors, and IGF-1, which may lead to psoriatic epidermal hyperplasia and therefore it is considered one of the causes of psoriasis [32]. The receptor IGF1R receives the IGF1 and activates the downstream pathways, P13K/AKT, and the well-known estrogen-mediated Ras/Raf/MEK/ERK pathway [33]. The phosphatidylinositol 3-kinase (PI3K) and protein kinase B (AKT) signaling pathway play a central role in multiple cellular functions, such as cell proliferation and survival. In psoriasis, the PI3K/AKT signaling pathway is crucial in regulating various cellular processes including cell survival and proliferation. One of its key effects is the inhibition of cell proliferation through the negative regulation of TF FOXO. In keratinocytes, PI3K signaling pathway promotes cell proliferation by activating AKT and other targets, and this is accomplished by decreasing the expression of FOXO [34]. TF FOXO affects two genes, *BCL6* and *IL7R*. Genetic polymorphisms in *IL7R* are associated with susceptibility to various autoimmune diseases. Studies have suggested that *IL7R* is involved in the pathogenesis of psoriasis [35,36]. The inactivation of Foxo1 leads to the release of the target gene *Bcl6* expression [37]. The B-cell lymphoma 6 (*Bcl6*) is selectively expressed by TFH cells, which affects the regulation of B-cell-mediated humoral immunity and thus affects the development of psoriatic disease [38,39].

In addition to affecting FOXO, AKT also phosphorylates the downstream protein IKBKB, which in turn phosphorylates the protein NFKBIA and causes its ubiquitination and degradation. After NFKBIA is phosphorylated, it initiates TF RELA to phosphorylate and transport to TF NF-kB. Nuclear factor kappa B (NF-κB) is a protein transcription factor that coordinates inflammation and other complex biological processes. It is a key regulatory element of multiple immune and inflammatory pathways, cell proliferation and differentiation, and apoptosis. Therefore, many studies believe that TF NF-κB is a key mediator involved in the pathogenesis of psoriasis [40,41]. TF NF-κB plays an important role in the inflammatory response, and it affects many target genes, including *VEGF*, *IL6*, *CXCL*, *COX-2*, *TNF-α*, and *BCL-XL* [42,43]. Interleukin 6 (IL-6) is a cytokine. Highly expressed IL-6 was reported to directly promote epidermal hyperplasia in psoriatic epithelial cells [22]. The target gene, Chemokine ligand (*CXCL*), is involved in the pathogenesis of psoriasis. Studies have demonstrated that *CXCR3* and *CXCL10* are present in keratinocytes and dermal infiltrates of active psoriatic plaques; effective treatment of active plaques reduces *CXCL10* expression in plaques [44].

The third microenvironment factor is IL17A. In psoriasis, IL-17A mainly acts on non-hematopoietic cells, especially epithelial cells, and continues to participate in the protective immunity of marginal tissues. In the skin, IL-17A leads to increased proliferation and abnormal differentiation of keratinocytes and disrupts the skin barrier by downregulating the expression of molecules involved in keratinocyte differentiation, such as filaggrin [45,46]. IL-17A is received by the receptor IL-17RC to activate the signaling cascade ACT1/TRAF6/MAP3K7/IKBKB [47]. In addition, the signaling protein TRAF6 also upregulates TF CEBPB. CEBPB family members are involved in epidermal keratinocyte differentiation. The TF CEBPB is an additional transcriptional regulator of IL-17A. Both NF-κB and CEBPB binding sites are overexpressed in the promoters of target genes *IL-17R* in psoriasis [48]. After ACT1 ubiquitination traf6, TF CEBPB will regulate target genes *IL6*, *CXCL*, *COX-2*, and *TNF-α*. The target gene *TNF-α* has multiple effects ranging from inflammation to apoptosis. Microenvironment factor IL-17A induces miR-378a expression in primary keratinocytes through NF-κB and CEBPB. Induction of miR-378a leads to the suppression of TF NFKBIA/IκBα, which allows for activation of the NF-κB pathway, leading to further induction of inflammatory mediators, such as *CXCL8* [49].

Another microenvironment factor is PGE2. Prostaglandin E 2 (PGE 2) is a major mediator of inflammatory disease and is produced by virtually all cells in the body. PGE2 is increased in the epidermis of psoriatic lesions and is associated with pruritus, and patients with psoriasis may experience pruritus secondary to PGE2 vasodilation and a decreased pruritus threshold [50,51,52]. After PGE2 is produced by the cell, the receptor EP1-4 receives it and begins to affect the protein Gby. It then affects both P13k/AKt/IKBKB/NF-κB signaling pathway and the MAPK signaling pathway Ras/Raf/MEK/ERK.

Tumor necrosis factor- α (TNF-α) is a pro-inflammatory cytokine that coordinates tissue homeostasis by regulating cytokine production, cell survival, and cell death. It is a key cytokine in the innate immune response and is increased in psoriatic lesions. Many studies considered it as the key cytokine to psoriasis [53,54,55]. The receptor TNFR1 receives signal TNF-α in the microenvironment and affects downstream protein RIP1, which is a key upstream regulator controlling the inflammatory signaling and the activation of multiple cell death signaling pathways, including apoptosis and necroptosis. RIP1 regulates the NF-κB-mediated inflammatory response along the signaling pathway MAP3K7/IKBKB in response to TNF stimulation [56]. In addition, TNF-α is also received by the EGF receptor (EGFR), which triggers the phosphorylation of downstream protein SRC and enters the P13K/AKT signaling pathway. In addition, EGFR also activates another signaling protein, GRb2, and enters the well-known estrogen-mediated Ras/Raf/MEK/ERK signaling pathway. Many experiments have established that mitogen-activated protein kinases p38 (p38 MAPKs) and extracellular signal-regulated kinase 1/2 (ERK1/2) are involved in the pathogenesis of psoriasis. They control several important cellular functions in cells, such as cell proliferation, differentiation, gene expression, and apoptosis [57,58,59]. Finally, TF ERK will regulate the target genes *c-jun* and *VEGF*. *C-jun* has been reported to be involved in proliferation, apoptosis, survival, tumorigenesis, and tissue morphogenesis. Furthermore, overexpression of TF ERK and gene *c-jun* are associated with hyperproliferation and abnormal differentiation of the psoriatic epidermis [60,61].

### 2.3. The Specific Molecular Pathogenesis of Non-Psoriasis

In tissues without psoriasis, we found that Wnt/β-catenin signaling regulated by the Hippo pathway plays an important role in anti-cell proliferation. Wnt ligands bind coreceptors of the Frizzled family and the LRP/arrow family. This binding activates intracellular Dishevelled (Dvl), which in turn regulates the activity of the serine-threonine protein kinase glycogen synthase kinase 3β (GSK-3β). The activation of Dvl inhibits GSK-3β in this complex, resulting in the stabilization and accumulation of β-catenin in the cytoplasm. After nuclear translocation, β-catenin interacts with members of the TCF/LEF family of DNA-binding molecules to affect target gene *Axin2* expression [62,63]. *Axin2* plays an important role in regulating β-catenin stability in the Wnt signaling pathway. The study pointed out that the expression of gene *Axin2* in normal tissue was significantly higher than that of psoriasis tissue [64].

### 2.4. Using Systems Drug Discovery and Design to Identify Potential Molecular Drugs to Combine as Multi-Molecule Drug for Psoriasis by the Prediction of Deep Neural Network-Based Drug-Target Interaction Model and Drug Design Specifications

By investigating the significant pathogenic mechanism of psoriasis based on core signaling pathways and their downstream cellular dysfunctions in the previous section, we selected NF-κB, STAT3, FOXO1, CEBPB, and ERK1/2 in Table 4 as significant biomarkers of pathogenic mechanism for psoriasis. Then according to the pharmacological properties of the molecular drugs, including regulatory ability, sensitivity, and toxicity as a drug design specification, some suitable potential drugs are discovered and designed to reverse the expression level of significant biomarkers.

To develop a systematic drug design and discovery process, as shown in Figure 5, we first pre-trained a deep neural network (DNN)-based DTI model by DTI databases in Section 4.6 to predict potential molecular drugs to target the identified biomarkers (drug targets) of psoriasis. The trained DNN-based DTI model allowed us to efficiently estimate the probability of interaction between candidate drugs and identified drug targets (biomarkers) for psoriasis in Table 5. We then filtered the candidate drugs in Table 6 based on their pharmacological properties, such as regulatory ability, sensitivity, and toxicity as a drug design specification, to obtain potential molecular drugs in Table 4, which are combined to form a multi-molecule drug for the treatment of psoriasis.

As shown in Figure 5, the training dataset consisted of 80,291 proven and 100,024 unproven drug-target interactions. To avoid imbalanced class distribution issues, we randomly selected an equal number of both types of interactions from the DTI database, as shown in Section 4.6. We also standardized the features in Equations (36) and (37) and reduced the dimensionality using principal component analysis (PCA) to obtain 996 of the 1359 features before training the DNN-based DTI model because the input layer of DNN we employed is only 996 nodes. The DNN-based DTI model consisted of an input layer with 996 nodes, four hidden layers with 512, 256, 128, and 64 neurons, and an output layer with one node, using ReLU activation for the neurons in hidden layers and sigmoid activation for the output layer. Dropout layers were also added to prevent overfitting. The learning curve of accuracy and loss during the DNN-based DTI model training process is shown in Figure 6 and Figure 7, respectively. The model achieved an average test accuracy of 98.2% with a standard deviation of 0.142% in five-fold cross-validation, as shown in Table 5. We also evaluated the model using ROC curves, with an AUC of 0.982, indicating significantly better performance than random prediction (AUC = 0.5), as shown in Figure 8. Finally, we used the best-performing DNN-based DTI model to predict candidate drugs based on the interaction probabilities with our identified drug targets in Table 6.

In order to identify viable candidate drugs, we utilize predictions based on the likelihood of the candidate drug binding (docking) to the chosen biomarkers. However, it is important to maintain a balance between drug efficacy and potential adverse effects, as potent molecular drugs can often come with a higher risk of harm. To ensure the safety and efficacy of the candidate molecular drugs predicted by the DNN-based DTI model, we incorporated some pharmacological properties of drug design specifications, including regulation ability, sensitivity, and toxicity. To evaluate the molecular drug regulation ability, we used the LINCS L1000 Level 5 dataset, which contains data on 12,328 genes treated with 19,811 small molecule compounds across 76 different human cell lines [65,66]. Positive values in the accommodation ability data represent an increase in expression levels, while negative values represent a decrease in expression levels. We used this dataset to identify potential molecular drugs from the candidate drugs that can restore the drug targets (biomarkers) to their normal expression levels. For drug sensitivity, we utilized the primary PRISM repurposing dataset, which includes information on 4518 compounds across 578 human cell lines [67]. We selected compounds with sensitivity values close to zero, indicating that the cell line is insensitive to chemical perturbations. In addition, we considered drug toxicity (LC50) using the ADMETlab 2.0 tool [68]. A higher LC50 value indicates lower toxicity and fewer side effects. Table 6 presents several candidate drugs that were predicted by the DNN-based DTI model for the identified biomarkers, along with their pharmacological information on regulatory ability, toxicity, and sensitivity. Based on the drug design specifications of suitable regulation ability, less sensitivity close to 0, and low toxicity, potential molecular drugs such as Naringin, Butein, and Betulinic-acid were selected. These molecular drugs were then combined as a multiple-molecule drug for the therapeutic treatment of psoriasis, as shown in Table 4.

## 3. Discussion

In this study, we aimed to discover a potential multi-molecule drug for the therapeutic treatment of psoriasis. To achieve this, we utilized core GWGENs by genome microarray data via the system identification and PNP method and then employed KEGG annotations to construct the core signaling pathways and to identify the biomarkers of pathogenesis as drug targets for the therapeutic treatment of psoriasis. We then employed DNN-based DTI model prediction and drug design specification to find potential molecular drugs to combine as a multi-molecule drug that possesses the proper toxicity, sensitivity, and regulation ability. With these system medicine approaches, consequentially, we were able to predict Butein, Naringin, and Betulinic-acid to combine as a potential multi-molecule drug for the therapeutic treatment of psoriasis.

Betulinic acid, or 3β-hydroxy-lup-20(29)-en-28-oic acid, is a naturally occurring pentacyclic lupane-type triterpenoid widely distributed in plants [69]. Previous studies used various experimental models to demonstrate the anti-inflammatory activity of betulinic acid [70]. Betulinic acid can inhibit the expression of NF-κB by reducing the activation of IKKβ, which is an inhibitor of TF NF-κB, and by decreasing the phosphorylation of IκBα, another inhibitor of NF-κB [71]. Furthermore, Betulinic acid acts as a natural inhibitor of the CEBP family [72]. Betulinic acid has been reported to inhibit TF ERK1/2, affecting *COX-2* expression [73]. In addition to the direct effects, other studies have reported that Betulinic acid can indirectly lower TF STAT3 and affect the regulation of gene *VEGF* [73]. For the treatment of psoriasis, although there are no clinical trials yet, the studies have confirmed that betulinic acid ameliorates psoriasis-like murine skin inflammation.

Butein (3,4,20,40-tetrahydroxychalcone) is present in numerous plants, including the stem-bark of cashews, herbs, such as caragana jubata, and rhus verniciflua, as well as the heartwood of Dalbergia odorifera. These plants have been used as herbal medicines for cancer treatment in many Asian countries, and the anti-cancer effects of these plant extracts are well-established [74]. Butein has been shown to exert various biological activities, such as antioxidant, anti-inflammatory, and anti-tumor activities [75]. For regulation on biomarkers, previous reports demonstrated that Butein induces anti-proliferative or pro-apoptotic effects in hepatic cells by downregulating signal transducer and activator of transcription-3 (STAT-3)-related gene expression and stimulating mitochondria-dependent caspase-3 activation [76]. In addition, studies have pointed out that in the human keratinocyte cell line, Butein inhibits *TNF-α*-induced expression of pro-inflammatory mediators by inhibiting NF-κB activation. In terms of treatment, Butein has immunomodulatory activity by inhibiting the expression of pro-inflammatory mediators in keratinocytes and is considered a therapeutic agent for the treatment of inflammatory skin diseases [77]. Butein has been shown to be a promising therapeutic agent for preventing adipose tissue inflammation and obesity-linked insulin resistance [78].

Naringin, chemically 4,5,7-trihydroxyflavanone-7-rhamnoglucoside, is a major flavanone glycoside obtained from tomatoes, grapefruits, and many other citrus fruits. A wide spectrum of beneficial effects has been attributed to Naringin, including cardiovascular, hypolipidemic, anti-inflammation, antidiabetic, neuroprotective, hepatoprotective, and anti-cancer activities [79,80]. For the regulation of biomarkers (drug targets), a previous study demonstrated that naringenin inhibits oxidative stress and inflammation, rescuing neuronal cell death. The mechanism was involved in the promotion of the SIRT1/FOXO1 signaling pathway. It has also been reported that Naringin stimulated the mitochondrial biogenesis pathway through regulation of the LKB1/AMPK/PGC-1α signaling pathway and upregulated FOXO1-mediated autophagy [81]. Naringin has been demonstrated to exert its anti-inflammatory effects by inhibiting the secretion, as well as inhibiting the phosphorylation of ERK1/2, JNK, and p38 MAPK, by blocking the activation of the NF-κB and MAPK signaling pathways [82]. In terms of therapeutic treatment, much research has demonstrated that Naringin exerts an anti-inflammatory effect on numerous chronic inflammatory diseases, including chronic bronchitis and inflammatory bowel disease [83]. In addition, Naringin combined with sericin significantly decreased the expression of mRNA and the production of all pro-inflammatory cytokines in hPBMCs from patients with psoriasis [84].

## 4. Materials and Methods

### 4.1. General Review of Constructing Core Genome-Wide Genetic and Epigenetic Networks (GWGENs) of Psoriasis and Non-Psoriasis

In order to construct the core genome-wide genetic and epigenetic networks (core GWGENs) for psoriasis and non-psoriasis, we first divided the data into a disease group and a healthy control group from the GSE117468 dataset. We then followed a four-step process, as shown in Figure 1. In the first step, we used a big database mining approach to construct a candidate PPIN (protein–protein interaction network) and a candidate GRN (gene regulatory network), which included the regulations of genes, miRNAs, and lncRNAs. In the second step, we identified the real GWGENs of psoriasis and non-psoriasis by using the microarray data of psoriasis and non-psoriasis to solve the constrained linear least squares estimation problems for the interaction parameters of PPIN and the regulation parameters of GRN using the system identification method. We pruned the false positives in the candidate GWGEN using the AIC method to obtain the real GWGENs of psoriasis and non-psoriasis [18]. In the third step, we extracted the core GWGENs by using the principal network projection PNP approach, which involves computing a projection value for each node in the real GWGENs to identify the top 6000 nodes, which is the largest number of nodes allowed for annotation by KEGG pathways with the higher projection values based on singular value to principal network structures with 85% energy of real GWGENs as the core GWGENs [85]. In the fourth step, we annotated the KEGG pathways of core GWGENs of psoriasis and non-psoriasis to construct their respective core signaling pathways. We then compared the upstream micro-environmental factors, core signaling pathways, and their corresponding downstream abnormal cellular functions of psoriasis and non-psoriasis to investigate the molecular mechanisms of carcinogenesis of psoriasis. Finally, we will identify significant biomarkers in core signaling pathways in response to cellular dysfunctions in pathogenic mechanism of psoriasis.

### 4.2. Data Preprocessing for Constructing the Candidate GWGEN

In this study, we obtained the dataset with accession number GSE117468 from NCBI, which contains microarray data of both psoriasis and non-psoriasis. We divided the microarray dataset into two groups: a disease group and a healthy control group. The dataset samples contain 128 samples of psoriasis and 128 samples of non-psoriasis. The dataset consists of mRNA, miRNA, and lncRNA data. To construct the candidate GWGEN, we used a binary matrix with a value of 1 assigned to nodes that showed interactions or regulation and a value of 0 assigned to nodes that did not. For the construction of the candidate PPIN, we referred to various databases, including DIP [86], IntAct [87], BioGRID [88], and MINT [89]. Similarly, we used several databases, such as HTRIdb [90], ITFP [91], TRANSFAC [92], CircuitDB [93], TargetScanHuman [94], and StarBase 2.0 [95] to construct the candidate GRN.

### 4.3. Building the Stochastic System Models of Candidate GWGEN to Identify Real GWGENs of Psoriasis and Non-Psoriasis

Based on the database and collected microarray data, we established candidate GWGEN. Next, we needed to identify the real GWGENs of psoriasis and non-psoriasis by their microarray data. Here, we constructed stochastic interaction and regulation models of candidate GWGEN, including protein–protein interactions, transcriptional regulations, miRNA and lncRNA regulations, as well as the basal level and stochastic noise due to model residuals and data measurement noise.

First, we established the following interaction equations between the a-th protein and its interacting proteins in the candidate PPIN of candidate GWGEN [96].
(1)$$p_{a}[n]=\sum_{r=1}^{R_{a}}\tau_{ar}p_{a}[n]p_{r}[n]+\sigma_{a,PPIN}+\lambda_{a,PPIN}[n]\;\mathrm{for}a=1,2\ldots ,A,n=1,2\ldots ,N$$
where p_a_ [n] and p_r_[n] indicate the expression level of the a-th and the r-th protein in the n-th sample; τ_ar_ represents the interaction ability between the a-th protein and the r-th protein; R_a_ stands for the total number of proteins interacting with the a-th protein; A is the total number of proteins in the candidate PPIN; N is the total number of the data samples (patients); $\sigma_{a,PPIM}$ represents the basal level of the a-th protein expression due to unknown interactions of histone modifications, such as phosphorylation and acetylation; $\lambda_{a,PPIN}$[n] indicates the stochastic noise of the a-th protein in the n-th sample because of data measurement noise.

After the protein interaction equations are completed, we need to establish the gene regulation equations. For gene regulation, the expression levels of TFs, lncRNAs, and miRNAs all have a great impact. For the gene regulatory model, the transcriptional regulation of the b-th gene in the n-th sample is described by the following Equation:(2)$$g_{b}[n]=\sum_{h=1}^{H_{b}}\alpha_{bh}t_{h}[n]+\sum_{i=1}^{I_{b}}\beta_{bi}l_{i}[n]-\sum_{j=1}^{J_{b}}\delta_{bj}m_{j}[n]g_{b}[n]+\sigma_{b}+\lambda_{b}[n]\;\mathrm{for}b=1,2\ldots ,B,n=1,2\ldots ,N$$
where g_b_[n], t_h_[n], l_i_[n], and m_j_[n] denote the expression level of the b-th gene, the h-th TF, the i-th lncRNA, and the j-th miRNA for the n-th sample; $\alpha_{bh}$ and $\beta_{bi}$ respectively indicate the transcriptional regulatory ability of the h-th TF and the i-th lncRNA on the b-th gene; $\delta_{bj}$ > 0 is the post-transcriptional regulatory ability of the j-th miRNA to inhibit the b-th gene; $H_{b}$, $I_{b}$ and $J_{b}$ represent the total binding number of TFs, lncRNAs and miRNAs of the regulation on the b-th gene, respectively; B is the total number of genes in candidate GRN; N is the total number of the data samples (patients); σ_b_ is the basal level of the b-th gene expression caused by unknown gene regulations, such as methylation; λ_b_[n] is the stochastic noise of the b-th gene in the n-th sample owing to model uncertainty and data noise.

Next, we established the lncRNA regulation Equation. The expression levels of TFs, lncRNAs, and miRNAs all have a great impact on the expression level of lncRNA. For the lncRNA regulatory model, the transcriptional regulation of the c-th lncRNA in the n-th sample is described by the following Equation [17]:(3)$$l_{c}[n]=\sum_{h=1}^{H_{c}}\varepsilon_{ch}t_{h}[n]+\sum_{i=1}^{I_{c}}\kappa_{ci}l_{i}[n]-\sum_{j=1}^{J_{c}}\gamma_{cj}m_{j}[n]l_{c}[n]+\sigma_{c}+\lambda_{c}[n]\;\mathrm{for}c=1,2\ldots ,C,n=1,2\ldots ,N$$
where $l_{c}$[n], t_h_[n], l_i_[n], and m_j_[n] denote the expression level of the c-th lncRNA, the h-th TF, the i-th lncRNA, and the j-th miRNA in the n-th sample, respectively; $\varepsilon_{ch}$ and $\kappa_{ci}$ separately indicate the transcriptional regulatory ability of the h-th TF and the i-th lncRNA on the c-th lncRNA; $\gamma_{cj}$ ≥ 0 is the post-transcriptional regulatory ability of the c-th miRNA to inhibit the c-th lncRNA; H_c_, I_c,_ and J_c_ individually represent the total binding number of TFs, lncRNAs and miRNAs on the c-th lncRNA; C is the total number of lncRNA; N is the total number of the data samples (patients); σ_c_ is the basal level of the c-th lncRNA expression caused by unknown regulations, such as methylation; λ_c_[n] is the stochastic noise of the c-th lncRNA in the n-th sample owing to model uncertainty and data noise.

Similarly, the expression levels of miRNAs are also regulated by TFs, lncRNAs, and other miRNAs. The regulatory model of miRNAs in candidate GWGEN is described in the following Equation:(4)$$m_{d}[n]=\sum_{h=1}^{H_{d}}x_{dh}t_{h}[n]+\sum_{i=1}^{I_{d}}y_{di}l_{i}[n]-\sum_{j=1}^{J_{d}}z_{dj}m_{j}[n]m_{d}[n]+\sigma_{d}+\lambda_{d}[n]\;\mathrm{for}d=1,2\ldots ,D,n=1,2\ldots ,N$$
where $m_{d}[n]$, t_h_[n], l_i_[n], and m_j_[n] denote the expression level of the d-th miRNA, the h-th TF, the i-th lncRNA and the j-th miRNA in the n-th sample, sequentially; x_dh_ and y_di_ separately indicate the transcriptional regulatory ability of the h-th TF and the i-th lncRNA on the d-th lncRNA, respectively; z_dj_ ≥ 0 is the post-transcriptional regulatory ability of the d-th miRNA to inhibit the d-th miRNA; H_d_, I_d,_ and J_d_ individually represent the total binding number of TFs, lncRNAs and miRNAs on the d-th miRNA; D is the total number of miRNA; N is the total number of the data samples (patients); σ_d_ is the basal level of the d-th lncRNA expression caused by unknown regulations; λ_d_[n] is the stochastic noise of the d-th miRNA in the n-th sample owing to model uncertainty and data noise.

### 4.4. Using the System Identification Scheme and System Order Detection Method to Prune False Positives of the Candidate GWGEN and Identify Real GWGENs of Psoriasis and Non-Psoriasis

According to the stochastic interaction and regulatory model, we constructed four models for candidate GWGENs above, including the candidate PPI model [96], gene regulatory model, lncRNA regulatory model, and miRNA regulatory model in Equations (1)–(4), respectively. Then, we used system identification and system order detection method to prune the false positive interactions in candidate GWGENs and obtain the real GWGENs of psoriasis and non-psoriasis by their microarray data. To estimate the interactive and regulative parameters, we needed to rewrite Equations (1)–(4) into the following linear regression form [97]:(5)$$\begin{array}{l} p_{i}\left[n\right]=\left[\begin{array}{ccc} \begin{array}{cc} p_{a}[n]p_{1}[n] & p_{a}[n]p_{2}[n] \end{array} & \ldots & \begin{array}{cc} p_{a}[n]p_{R_{a}}[n] & 1 \end{array} \end{array}\right]\times \left[\begin{array}{c} \tau_{a1}\\ \tau_{a1}\\ \vdots \\ \tau_{aR_{a}}\\ \sigma_{a,PPIN} \end{array}\right]+\lambda_{a,PPIN}[n]\\ ≜\phi_{a}[n]\cdot \theta_{a}+\lambda_{a,PPIN}[n],\;\mathrm{for}\;a=1,2,\cdots ,A,\;n=1,\ldots ,N \end{array}$$
(6)$$\begin{array}{l} g_{b}\left[n\right]=\left[\begin{array}{cccc} t_{1}[n] & \ldots & t_{H_{b}}[n] & \begin{array}{cccc} w_{1}[n] & \ldots & w_{I_{b}}[n] & \begin{array}{cccc} s_{1}[n]g_{b}[n] & \ldots & s_{J_{b}}[n]g_{b}[n] & 1 \end{array} \end{array} \end{array}\right]\times \left[\begin{array}{c} \alpha_{b1}\\ \vdots \\ \alpha_{bH_{b}}\\ \beta_{b1}\\ \begin{array}{c} \vdots \\ \beta_{bI_{b}}\\ \begin{array}{c} -\delta_{b1}\\ \vdots \\ -\delta_{bJ_{b}}\\ \sigma_{b} \end{array} \end{array} \end{array}\right]+\lambda_{b}[n]\\ ≜\phi_{b}[n]\cdot \theta_{b}+\lambda_{b}[n],\;\mathrm{for}\;b=1,2,\cdots ,B,\;n=1,\ldots ,N \end{array}$$
(7)$$\begin{array}{l} l_{c}[n]=\left[\begin{array}{cccc} t_{1}[n] & \ldots & t_{H_{c}}[n] & \begin{array}{cccc} w_{1}[n] & \ldots & w_{I_{c}}[n] & \begin{array}{cccc} s_{1}[n]g_{b}[n] & \ldots & s_{J_{c}}[n]g_{c}[n] & 1 \end{array} \end{array} \end{array}\right]\times \left[\begin{array}{c} \varepsilon_{c1}\\ \vdots \\ \varepsilon_{cH_{c}}\\ \kappa_{c1}\\ \begin{array}{c} \vdots \\ \kappa_{I_{c}}\\ \begin{array}{c} -\gamma_{c1}\\ \vdots \\ -\gamma_{cJ_{c}}\\ \sigma_{c} \end{array} \end{array} \end{array}\right]+\lambda_{c}[n]\\ ≜\phi_{c}[n]\cdot \theta_{c}+\lambda_{c}[n],\;\mathrm{for}\;c=1,2,\cdots ,C,\;n=1,\ldots ,N \end{array}$$
(8)$$\begin{array}{l} m_{d}[n]=\left[\begin{array}{cccc} t_{1}[n] & \ldots & t_{H_{d}}[n] & \begin{array}{cccc} w_{1}[n] & \ldots & w_{I_{d}}[n] & \begin{array}{cccc} s_{1}[n]g_{d}[n] & \ldots & s_{J_{d}}[n]g_{d}[n] & 1 \end{array} \end{array} \end{array}\right]\times \left[\begin{array}{c} x_{d1}\\ \vdots \\ x_{dH_{d}}\\ y_{d1}\\ \begin{array}{c} \vdots \\ y_{dI_{d}}\\ \begin{array}{c} -z_{d1}\\ \vdots \\ -z_{dJ_{d}}\\ \sigma_{d} \end{array} \end{array} \end{array}\right]+\lambda_{d}[n]\\ ≜\phi_{d}[n]\cdot \theta_{d}+\lambda_{d}[n],\;\mathrm{for}\;d=1,2,\cdots ,D,\;n=1,\ldots ,N \end{array}$$
where ϕ_a_[n], ϕ_b_[n], ϕ_c_[n], ϕ_d_[n] are regression vectors of the expression data of the a-th protein, the b-th gene, the c-th lncRNA, and the d-th miRNA in the n-th sample, respectively; θ_a_ denotes the parameter vector of protein–protein interactions and the basal level of the a-th proteins; θ_b_, θ_c_, θ_d_ indicate the parameter vector of the transcriptional regulatory ability and basal level of the b-th gene, the c-th lncRNA, and the d-th miRNA, respectively; λ_a_[n],λ_b_[n],λ_c_[n],λ_d_[n] are the stochastic noise due to model residue and data noise of the a-th protein, the b-th gene, the c-th lncRNA, and the d-th miRNA for n samples, respectively.

Then, linear Equations (5)–(8) can be further rewritten with N microarray data samples as the following Equations:(9)$$\left[\begin{array}{c} p_{a}[1]\\ p_{a}[2]\\ \vdots \\ p_{a}[N] \end{array}\right]=\left[\begin{array}{c} \phi_{a}[1]\\ \phi_{a}[1]\\ \vdots \\ \phi_{a}[N] \end{array}\right]\theta_{a,P}+\left[\begin{array}{c} \lambda_{a}[1]\\ \lambda_{a}[2]\\ \vdots \\ \lambda_{a}[N] \end{array}\right],\;\mathrm{for}\;a=1,2,\cdots ,A$$
(10)$$\left[\begin{array}{c} g_{b}[1]\\ g_{b}[2]\\ \vdots \\ g_{b}[N] \end{array}\right]=\left[\begin{array}{c} \phi_{b}[1]\\ \phi_{b}[2]\\ \vdots \\ \phi_{b}[N] \end{array}\right]\theta_{b,G}+\left[\begin{array}{c} \lambda_{b}[1]\\ \lambda_{b}[2]\\ \vdots \\ \lambda_{b}[N] \end{array}\right],\;\mathrm{for}\;b=1,2,\cdots ,B$$
(11)$$\left[\begin{array}{c} l_{c}[1]\\ l_{c}[2]\\ \vdots \\ l_{c}[N] \end{array}\right]=\left[\begin{array}{c} \phi_{c}[1]\\ \phi_{c}[2]\\ \vdots \\ \phi_{c}[N] \end{array}\right]\theta_{c,G}+\left[\begin{array}{c} \lambda_{c}[1]\\ \lambda_{c}[2]\\ \vdots \\ \lambda_{c}[N] \end{array}\right],\;\mathrm{for}\;c=1,2,\cdots ,C$$
(12)$$\left[\begin{array}{c} m_{d}[1]\\ m_{d}[2]\\ \vdots \\ m_{d}[N] \end{array}\right]=\left[\begin{array}{c} \phi_{d}[1]\\ \phi_{d}[2]\\ \vdots \\ \phi_{d}[N] \end{array}\right]\theta_{d,M}+\left[\begin{array}{c} \lambda_{d}[1]\\ \lambda_{d}[2]\\ \vdots \\ \lambda_{d}[N] \end{array}\right],\;\mathrm{for}\;d=1,2,\cdots ,D$$

The Equations (9)–(12) can be simply expressed as the following equations, respectively:(13)$$P_{a}=\Phi_{a}\cdot \Theta_{a}+\Omega_{a},\;\mathrm{for}\;a=1,2,\cdots ,A$$
(14)$$G_{b}=\Phi_{b}\cdot \Theta_{b}+\Omega_{b},\;\mathrm{for}\;b=1,2,\cdots ,B$$
(15)$$L_{c}=\Phi_{c}\cdot \Theta_{c}+\Omega_{c},\;\mathrm{for}\;c=1,2,\cdots ,C$$
(16)$$M_{d}=\Phi_{d}\cdot \Theta_{d}+\Omega_{d},\;\mathrm{for}\;d=1,2,\cdots ,D$$

Then, we can estimate the parameter vectors Θ_a_, Θ_b_, Θ_c_, Θ_d_ by their corresponding microarray data of N samples (patients). If the component number of the parameter vector of protein in PPIN and genes in GRN is larger than half (N/2) of the dataset samples, it may cause an overfitting problem in the parameter estimation. Therefore, we estimated the parameter vector by solving the constrained linear least-squares parameter estimation problem as follows [97]:(17)$${\hat{\Theta }}_{a}=\underset{\Theta_{a}}{\arg\min}\frac{1}{2}{\left\|\Phi_{a}\cdot \Theta_{a}-P_{a}\right\|}_{2}^{2}$$
(18)$$\begin{array}{l} {\hat{\Theta }}_{b}=\underset{\Theta_{b}}{\arg\min}\frac{1}{2}{\left\|\Phi_{b}\cdot \Theta_{b}-G_{b}\right\|}_{2}^{2},\\ subject\;to\;\left[\left.\underset{H_{b}}{\underbrace{\begin{array}{cccc} 0 & \cdots & \cdots & 0\\ \vdots & \ddots & & \vdots \\ \vdots & & \ddots & \vdots \\ 0 & \cdots & \cdots & 0 \end{array}}}\right|\left.\underset{I_{b}}{\underbrace{\begin{array}{cccc} 0 & \cdots & \cdots & 0\\ \vdots & \ddots & & \vdots \\ \vdots & & \ddots & \vdots \\ 0 & \cdots & \cdots & 0 \end{array}}}\right|\left.\underset{J_{b}}{\underbrace{\begin{array}{cccc} 1 & \cdots & \cdots & 0\\ \vdots & \ddots & & \vdots \\ \vdots & & \ddots & \vdots \\ 0 & \cdots & \cdots & 1 \end{array}}}\right|\begin{array}{c} 0\\ \vdots \\ \vdots \\ 0 \end{array}\right]\Theta_{b}\leq \left[\begin{array}{c} 0\\ \vdots \\ \vdots \\ 0 \end{array}\right] \end{array}$$
(19)$$\begin{array}{l} {\hat{\Theta }}_{c}=\underset{\Theta_{c}}{\arg\min}\frac{1}{2}{\left\|\Phi_{c}\cdot \Theta_{c}-L_{c}\right\|}_{2}^{2},\\ subject\;to\;\left[\left.\underset{H_{c}}{\underbrace{\begin{array}{cccc} 0 & \cdots & \cdots & 0\\ \vdots & \ddots & & \vdots \\ \vdots & & \ddots & \vdots \\ 0 & \cdots & \cdots & 0 \end{array}}}\right|\left.\underset{I_{c}}{\underbrace{\begin{array}{cccc} 0 & \cdots & \cdots & 0\\ \vdots & \ddots & & \vdots \\ \vdots & & \ddots & \vdots \\ 0 & \cdots & \cdots & 0 \end{array}}}\right|\left.\underset{J_{c}}{\underbrace{\begin{array}{cccc} 1 & \cdots & \cdots & 0\\ \vdots & \ddots & & \vdots \\ \vdots & & \ddots & \vdots \\ 0 & \cdots & \cdots & 1 \end{array}}}\right|\begin{array}{c} 0\\ \vdots \\ \vdots \\ 0 \end{array}\right]\Theta_{c}\leq \left[\begin{array}{c} 0\\ \vdots \\ \vdots \\ 0 \end{array}\right] \end{array}$$
(20)$$\begin{array}{l} {\hat{\Theta }}_{d}=\underset{\Theta_{d}}{\arg\min}\frac{1}{2}{\left\|\Phi_{d}\cdot \Theta_{d}-M_{d}\right\|}_{2}^{2},\\ subject\;to\;\left[\left.\underset{H_{d}}{\underbrace{\begin{array}{cccc} 0 & \cdots & \cdots & 0\\ \vdots & \ddots & & \vdots \\ \vdots & & \ddots & \vdots \\ 0 & \cdots & \cdots & 0 \end{array}}}\right|\left.\underset{I_{d}}{\underbrace{\begin{array}{cccc} 0 & \cdots & \cdots & 0\\ \vdots & \ddots & & \vdots \\ \vdots & & \ddots & \vdots \\ 0 & \cdots & \cdots & 0 \end{array}}}\right|\left.\underset{J_{d}}{\underbrace{\begin{array}{cccc} 1 & \cdots & \cdots & 0\\ \vdots & \ddots & & \vdots \\ \vdots & & \ddots & \vdots \\ 0 & \cdots & \cdots & 1 \end{array}}}\right|\begin{array}{c} 0\\ \vdots \\ \vdots \\ 0 \end{array}\right]\Theta_{d}\leq \left[\begin{array}{c} 0\\ \vdots \\ \vdots \\ 0 \end{array}\right] \end{array}$$

The constraints in Equations (18)–(20) ensure that the estimated post-transcriptional regulatory ability of miRNAs on genes, lncRNAs, and miRNAs is negative. Therefore, we can estimate the optimal vectors ${\hat{\Theta }}_{a}$,${\hat{\Theta }}_{b}$,${\hat{\Theta }}_{c}$,${\hat{\Theta }}_{d}$ of the protein interaction, gene, lncRNA, and miRNA regulation by solving the constrained least squares parameter estimation problems in Equations (17)–(20) with the MATLAB Optimization Toolbox, respectively.

After solving the constrained least squares parameter estimation problems above, we obtained the parameters of interactive ability, regulatory ability, and basal level for each protein, gene, miRNA, and lncRNA in candidate GWGENs of psoriasis and non-psoriasis by their microarray data. Then, we pruned false positives in candidate GWGENs to obtain real GWGENs of psoriasis and non-psoriasis through the system order detection method via the Akaike Information Criterion (AIC) method. The AIC method for detecting the order of protein, gene, lncRNA, and miRNA in candidate GWGENs is given as follows [96]:(21)$$\begin{array}{l} AIC(R_{a})=\log(\rho^{2}{}_{a})+\frac{2(R_{a}+1)}{N}\\ where\;\rho_{a}=\sqrt{\frac{{\left(P_{a}-\Phi_{a}\cdot {\hat{\Theta }}_{a}\right)}^{T}(P_{a}-\Phi_{a}\cdot {\hat{\Theta }}_{a})}{N}} \end{array}$$
and ρ_a_ is the estimated residual error of the a-th protein, which was calculated from the least square estimated parameter ${\hat{\Theta }}_{a}$ in Equation (17). R_a_ is the number of protein interactions with the a-th protein.
(22)$$\begin{array}{l} AIC(H_{b},I_{b},J_{b})=\log(\rho^{2}{}_{b})+\frac{2(H_{b}+I_{b}+J_{b}+1)}{N}\\ where\;\rho_{b}=\sqrt{\frac{{\left(G_{b}-\Phi_{b}\cdot {\hat{\Theta }}_{b}\right)}^{T}(G_{b}-\Phi_{b}\cdot {\hat{\Theta }}_{b})}{N}} \end{array}$$
and ρ_b_ is the estimated residual error of the b-th gene, which was calculated from the least square estimated parameter ${\hat{\Theta }}_{b}$ in the constrained estimation parameter problem Equation (18). H_b_, I_b,_ and J_b_ are the number of regulations of genes, lncRNAs, and miRNAs on the b-th gene.
(23)$$\begin{array}{l} AIC(H_{c},I_{c},J_{c})=\log(\rho^{2}{}_{c})+\frac{2(H_{c}+I_{c}+J_{c}+1)}{N}\\ where\;\rho_{c}=\sqrt{\frac{\left(G_{c}-{\left(\Phi_{c}\cdot {\hat{\Theta }}_{c}\right)}^{T}(G_{c}-(\Phi_{c}\cdot {\hat{\Theta }}_{c}\right)}{N}} \end{array}$$
and ρ_c_ is the estimated residual error of the c-th lncRNA, which was calculated from the least square estimated parameter ${\hat{\Theta }}_{c}$ in the constrained estimation parameter problem Equation (19). H_c_, I_c,_ and J_c_ are the number of regulations of genes, lncRNAs, and miRNAs on the c-th gene, respectively.
(24)$$\begin{array}{l} AIC(H_{d},I_{d},J_{d})=\log(\rho^{2}{}_{d})+\frac{2(H_{d}+I_{d}+J_{d}+1)}{N}\\ where\;\rho_{d}=\sqrt{\frac{\left(G_{d}-{\left(\Phi_{d}\cdot {\hat{\Theta }}_{d}\right)}^{T}(G_{d}-(\Phi_{d}\cdot {\hat{\Theta }}_{d}\right)}{N}} \end{array}$$
and ρ_d_ is the estimated residual error of the d-th miRNA, which be calculated from the least square estimated parameter ${\hat{\Theta }}_{d}$ in the constrained estimation parameter problem Equation (19). H_d_, I_d,_ and J_d_ are the number of regulations of genes, lncRNAs, and miRNAs on the d-th gene, respectively.

Akaike Information Criterion (AIC) is an index used to compare the pros and cons of different statistical models. The smaller the AIC value, the better the predictive ability of the model. To obtain the real GWGENs, we minimize the four AICs by equations as follows:(25)$$R_{a}^{*}=\underset{R_{a}}{\arg\min}AIC(R_{a})\;\mathrm{for}\;a=1,\ldots ,A$$
(26)$$H_{b}^{*},I_{b}^{*},J_{b}^{*}=\underset{H_{b},I_{b},J_{b}}{\arg\min}AIC(H_{b},I_{b},J_{b}),\;\mathrm{for}\;b=1,\ldots ,B$$
(27)$$H_{c}^{*},I_{c}^{*},J_{c}^{*}=\underset{H_{c},I_{c},J_{c}}{\arg\min}AIC(H_{c},I_{c},J_{c}),\;\mathrm{for}\;c=1,\ldots ,C$$
(28)$$H_{d}^{*},I_{d}^{*},J_{d}^{*}=\underset{H_{d},I_{d},J_{d}}{\arg\min}AIC(H_{d},I_{d},J_{d}),\;\mathrm{for}\;d=1,\ldots ,D$$
where R^*^_a_ means the real number of protein interactions with the a-th protein. H^*^_b_, I^*^_b_, and J^*^_b_ denote the real number of regulations of TFs, lncRNAs, and miRNAs on the b-th gene. H^*^_c_, I^*^_c_, and J^*^_c_ denote the real number of regulations of TFs, lncRNAs, and miRNAs on the c-th lncRNA. H^*^_d_, I^*^_d_, and J^*^_d_ denote the real number of regulations of TFs, lncRNAs, and miRNAs on the d-th miRNA. After solving the above AIC minimum problem, the false positives out of the real number of interactions and regulations must be removed from candidate GWGENs and obtain the real GWGENs of psoriasis and non-psoriasis in Figure 2.

### 4.5. Extracting the Core GWGENs from the Real GWGENs by the Principal Network Projection (PNP) Method

After system order detection and identification method by the corresponding microarray data, we successfully obtained real GWGENs of psoriasis and non-psoriasis in Figure 2. However, it is still too complicated to figure out the molecular mechanisms of psoriasis. Therefore, we needed to employ KEGG pathways to annotate signaling pathways of real GWGENs. However, at present, KEGG pathways can only annotate GWGENs with 6000 nodes at most. Therefore, we used the PNP method to extract 6000 nodes from real GWGENs of psoriasis and non-psoriasis for pathway annotation by KEGG. The PNP method is based on singular value decomposition (SVD) of real GWGENs in Figure 2 to extract k = 6000 important nodes in real GWGENs to construct core GWGENs in Figure 3. The procedure of the PNP method is given as follows: First, in order to employ the singular value decomposition technique, we constructed a network matrix W from real GWGENs.
(29)W=wprotien↔protien00wTF→genewlncRNA→genewmiRNA→genewTF→lncRNAwlncRNA→lncRNAwmiRNA→lncRNAwTF→miRNAwlncRNA→miRNAwmiRNA→miRNA
where wprotien↔protien is a sub-matrix which records the estimated protein interaction abilities of PPIN. The bidirectional arrow means the protein interaction is bidirectional; the sub-network matrices wTF→gene, wlncRNA→gene, and wmiRNA→gene record the estimated transcriptional regulation abilities of TFs, lncRNAs, and miRNAs on genes, respectively; wTF→lncRNA, wlncRNA→lncRNA, and wmiRNA→lncRNA indicate matrices of the estimated transcriptional regulation abilities of TFs, lncRNAs, and miRNAs on lncRNAs, respectively; wTF→miRNA, wlncRNA→miRNA, wmiRNA→miRNA are the matrices of the estimated transcriptional regulation abilities of TFs, lncRNAs and miRNAs on miRNAs, respectively. The following matrix is the detail on the network matrix of W of real GWGENs:(30)W=τ^11⋯τ^1r⋯τ^1Ra0⋯0⋯00⋯0⋯0⋮⋱⋮⋱⋮⋮⋱⋮⋱⋮⋮⋱⋮⋱⋮τ^a1⋯τ^ar⋯τ^aRa0⋯0⋯00⋯0⋯0⋮⋱⋮⋱⋮⋮⋱⋮⋱⋮⋮⋱⋮⋱⋮τ^A1⋯τ^Ar⋯τ^ARa0⋯0⋯00⋯0⋯0α11⋯α1h⋯α1Hbβ11⋯β1i⋯β1Ib−δ11⋯−δ1j⋯−δ1Jb⋮⋱⋮⋱⋮⋮⋱⋮⋱⋮⋮⋱⋮⋱⋮αb1⋯αbh⋯αbHbβb1⋯βbi⋯βbIb−δb1⋯−δbj⋯−δbJb⋮⋱⋮⋱⋮⋮⋱⋮⋱⋮⋮⋱⋮⋱⋮αB1⋯αBh⋯αBHbβB1⋯βBi⋯βBIb−δB1⋯−δBj⋯−δBJbε11⋯ε1h⋯ε1Hcκ11⋯κ1i⋯κ1Ic−γ11⋯−γ1j⋯−γ1Jc⋮⋱⋮⋱⋮⋮⋱⋮⋱⋮⋮⋱⋮⋱⋮εc1⋯εch⋯εcHcκc1⋯κci⋯κcIc−γc1⋯−γcj⋯−γcJc⋮⋱⋮⋱⋮⋮⋱⋮⋱⋮⋮⋱⋮⋱⋮εC1⋯εCh⋯εCHcκC1⋯κci⋯κCIc−γC1⋯−γCj⋯−γCJcx11⋯x1h⋯x1Hdy11⋯y1i⋯y1Id−z11⋯−z1j⋯−z1Jd⋮⋱⋮⋱⋮⋮⋱⋮⋱⋮⋮⋱⋮⋱⋮xd1⋯xdh⋯xdHdyd1⋯ydi⋯ydId−zd1⋯−zdj⋯−zdJd⋮⋱⋮⋱⋮⋮⋱⋮⋱⋮⋮⋱⋮⋱⋮xD1⋯xDh⋯xDHdyD1⋯yDi⋯yDId−zD1⋯−zDj⋯−zDJd∈ℝ(A*+B*+C*+D*)×(H*+I*+J*)

Then, we performed a singular value decomposition of the network matrix W as follows:(31)W=UΣVTU∈ℝ(A*+B*+C*+D*)×(A*+B*+C*+D*)Σ∈ℝ(A*+B*+C*+D*)×(H*+I*+J*)V∈ℝ(H*+I*+J*)×(H*+I*+J*)
(32)Σ=σ1⋯0⋯0⋮⋱⋮⋱⋮0⋯σk⋯0⋮⋱⋮⋱⋮0⋯0⋯σH*+I*+J*0⋯0⋯0⋮⋱⋮⋱⋮0⋯0⋯0∈ℝ(A*+B*+C*+D*)×(H*+I*+J*)

In order to extract the core GWEN from the real GWEN, we only retained the top K singular values of matrix Σ in Equation (32), which have at least 85% energy of real GWGENs. Similarly, the matrix U retains the top K rows, and V^T^ retains the top-K columns to construct a significant network structure with at least 85% energy as follows [18,96]:(33)$$E_{k}=\frac{\sum_{x=1}^{K}\sigma_{x}^{2}}{\sum_{m=1}^{H^{*}+I*+J*}\sigma_{m}^{2}}\geq 0.85$$

Next, we project each node of the real GWGENs (i.e., each row of network matrix W) separately to the top K principal singular vectors and apply the 2-norm projection value of each node such as protein, gene,miRNA, and lncRNA in the real GWGENs of psoriasis and non-psoriasis to the top K principal singular vectors as follows:(34)$$\begin{array}{l} proj(a,b)=w_{a}\cdot v_{b}{}^{T}\;\\ P(a)=\sqrt{\sum_{b=1}^{K}proj(a,b)}\;,\\ \mathrm{for}\;a=1,\ldots ,(A^{*}+B^{*}+C^{*}+D^{*})\;,\;b=1,\ldots ,K \end{array}$$
where proj(a,b) denotes the projection value of the a-th node to the b-th principal singular vector; w_a_ and v_b_ are the a-th row of the W vector and the b-th principal singular vector. P(a) represents the 2-norm projection value of the a-th node to the top K singular vectors. The larger the P(a), the more influence the a-th node of real GWGENs is on the network structure. Conversely, if P(a) is very close to zero, then the a-th node of real GWGENs is almost independent of the significant network structure.

The core GWGENs of psoriasis and non-psoriasis were identified, as shown in Figure 3, by extracting the top 6000 significant nodes based on their projection values P(a) in Equation (34). These core GWGENs were then annotated by KEGG pathways, which allowed us to identify the core signaling pathways of psoriasis and non-psoriasis in Figure 4. We compared these two core signaling pathways in Figure 4 to gain insights into the pathogenesis underlying the development of psoriasis. Based on the investigated pathogenic mechanism in Figure 4, we selected essential biomarkers as potential drug targets for the therapeutic treatment of psoriasis.

### 4.6. Design and Discovery of Multi-Molecule Drug through the Prediction of DNN-Based DTI Model for Treatment of Psoriasis by DNN-Based DTI Model

After finding the essential biomarkers as drug targets for the pathogenesis of psoriasis, we needed to identify the corresponding candidate molecular drugs to target these essential biomarkers. DNN-based DTI model has been effectively employed to predict candidate drugs to interact with significant biomarkers (drug targets) [96].

To train a DNN-based DTI model to predict the interactions of drug candidates with significant biomarkers of psoriasis, we integrated multiple DTI databases, including STITCH, BIDD, UniProt, DrugBank, ChEMBL, PubChem, and KEGG [98,99,100,101,102,103], which provide information about molecular interactions, drug and target features. Drug features include structure, topology, geometric descriptors, and other molecular properties, while the sequence of the target protein is representative because the complete target protein information is usually encoded in the sequence. Target features are calculated from the structural and physicochemical features of proteins and peptides in amino acid sequences. After converting the properties of drugs and targets into features, we used the Python package PyBioMed under the Python 3.7 environment [104], and each drug-target pair was represented as a feature vector by concatenating the corresponding feature vectors. The feature vector of the drug-target pair is as follows [105]:(35)$$x_{\mathrm{drug}-\mathrm{target}}=[D,T]=[d_{1},\ldots ,d_{M},t_{1},\ldots ,t_{N}]$$
where x_drug-target_ refers to the feature vector for the drug-target pair, while D and T represent the feature vectors for the drug and target, respectively, d_m_ and t_n_ refer to the m-th drug feature and n-th target feature, respectively, where M and N indicate the total number of drug and target features, respectively.

Before using the drug-target vector in Equation (35) as training data for a deep neural network (DNN) as a DTI model to predict candidate molecular drugs for significant biomarkers (drug targets) of psoriasis, we needed to pre-process the data to make it suitable as an input to the DNN. Because there were more negative data with unknown interactions than positive data with known interactions, it was necessary to downsample the negative data to reduce the imbalance problem between the numbers of positive and negative data. The purpose of downsampling is to randomly reduce the amount of negative data to the same number as positive data, allowing the model to learn equally from both types of training data. Furthermore, since the variables of the feature vectors in Equation (35) in each drug-target pair are measured in different units, we need to normalize each feature vector to normalize the importance among the feature vectors. In other words, the purpose of preprocessing the training data is to enable DNN to better learn the model of drug-target interaction. The mathematical formulas for the normalization of the features of the drug and target are as follows:(36)$$d_{m}^{*}=\frac{d_{m}-\mu_{m}}{\sigma_{m}}\;,\forall m=1,\ldots ,M$$
(37)$$t_{n}^{*}=\frac{t_{n}-\mu_{n}}{\sigma_{n}}\;,\forall n=1,\ldots ,N$$
where d_m_* represents the m-th drug feature represents the m-th drug feature after standardization, and d_m_ represents the original m-th drug feature. Similarly, t_n_* represents the n-th target feature after standardization, and t_n_ represents the original n-th target feature. The mean and standard deviation of each drug feature are represented by μ_m_ and σ_m_, respectively. Likewise, the mean and standard deviation of each target feature is represented by μ_n_ and σ_n_, respectively. Equations (36) and (37) refer to a standardization process for drug and target features, respectively.

Since the DNN, which is employed as a DTI model, has only 996 inputs, we needed to reduce the dimension of drug target features in the feature vectors in Equation (35) so that they could be input into the DNN. To reduce the input number of features to DNNs as the DTI model, we downsample the feature vectors of the drug-target pair to reduce their dimension using the principal component analysis (PCA) method. After data preprocessing, we use 75% of the data as training data and 25% of the data as testing data. The DNN-based DTI model in Figure 5 has four hidden layers containing 512, 256, 128, and 64 neurons, respectively. We used binary cross entropy as the loss function, the learning rate is 0.001, and we used the Adam algorithm as the optimizer. In addition, we set epochs to 100 and batch size to 100. Each hidden layer has a rectified linear unit (ReLU) activation function to avoid the vanishing gradient problem in deep learning. We also use the five-fold cross-validation method for learning data to check the stability and prediction performance of the DNN-based DTI model. We applied the early stopping approach to avoid the overfitting problem. The activation function of the output layer is a sigmoid function, which can limit the output value in the range of 0 to 1 as the probability predicted by the DNN-based DTI model. Because the drug-target interaction (DTI) is a binary classification issue, the cost function which we chose to calculate the model loss is the binary cross-entropy as follows [97]:(38)$$C_{n}(p_{n},{\hat{p}}_{n})=-[p_{n}\log{\hat{p}}_{n}+(1-p_{n})\log(1-{\hat{p}}_{n})]$$
(39)$$L(w,b)=\frac{1}{N}\sum_{n=1}^{N}C_{n}(w,b)$$

For the n-th sample, we have a true positive interaction probability p_n_ and a predicted positive interaction probability ${\hat{p}}_{n}$, and a true negative interaction probability 1 − p_n_ and a predicted negative interaction probability 1−${\hat{p}}_{n}$. We use binary cross-entropy as the loss function to compute the total loss $C_{n}(p_{n},{\hat{p}}_{n})$ by averaging the loss L(w,b) per sample. w denotes the weighting vector; b indicates the bias vector of DNN. N is the total number of training data.

We utilized the backward propagation algorithm [106] to update the parameter vector θ of the weight vector w and bias vector b based on the cost function. This involves calculating the gradient to obtain the optimal parameter vector θ* for the DNN-based DTI model as follows:(40)$$\theta =\left[\begin{array}{c} w\\ b \end{array}\right]$$
(41)$$\theta^{*}=\underset{\theta }{\arg\min}L(\theta )$$
(42)$$\begin{array}{l} \theta^{y}=\theta^{y-1}-\eta \nabla L(\theta^{y-1})\;,\\ \mathrm{where}\;\nabla L(\theta^{y-1})=\left[\begin{array}{c} \frac{\partial L(\theta^{y-1})}{\partial w}\\ \frac{\partial L(\theta^{y-1})}{\partial b} \end{array}\right] \end{array}$$
and y means the y-th interaction of the learning process by training of the y-th drug-target feature vector; η is the learning rate which is set as 0.001 in this study; $\nabla L(\theta^{y-1})$ means the gradient of $L(\theta^{y-1})$.

After training the DNN-based DTI model by feature vectors of drug-target pairs in Equation (35), we used the area under the curve (AUC) and receiver operating characteristic (ROC) curves in Figure 8 to evaluate the prediction performance of the model. For classification problems, this is one of the most useful evaluation metrics to visualize the performance of the DNN-based DTI model. The larger the area under the ROC curve, the higher the AUC and the higher the accuracy of the DNN-based DTI model in predicting true positive and true negative drug-target interactions. The mathematical formulas for AUC and ROC curves are as follows [96]:(43)$$TPR=\frac{TP}{TP+FN}$$
(44)$$\mathrm{TNR}=\frac{TN}{TN+FP}$$
(45)$$FPR=\frac{\mathrm{FP}}{\mathrm{TN}+\mathrm{FP}}=1-\mathrm{TNR}$$
where TP means the judgment is true and the fact is also true; TN means the judgment is false and real value is also false; FP means the judgment is true, but the real value is false; FN means the judgment is false, but the real value is true;

By using the DNN-based DTI model, we successfully obtained five candidate molecular drugs in Table 6, which can be pruned by drug design specifications to obtain potential molecular drugs for psoriasis. We screened out the potential molecular drugs from candidate molecular drugs in Table 6 based on some pharmacological properties such as adequate regulation ability, sensitivity, and toxicity as drug design specifications. Specifically, we referred to regulatory capacity data from the L1000 Level 5 dataset [65], from which molecular drugs were selected against selected biomarkers (drug targets). For example, if the gene expression of the selected biomarkers (drug targets) is abnormally upregulated, we will select the drug with a negative correlation. Conversely, if the gene expression of the selected biomarkers (drug targets) is abnormally downregulated, a drug with a positive correlation is selected. At the same time, we selected molecular drugs with sensitivity values close to zero corresponding datasets obtained from primary PRISM repurposing datasets [67]. A sensitivity value close to zero indicates that the cell is not sensitive to the molecular drug. In consideration of drug toxicity, we used the ADMETlab 2.0 tool to refer to the LC50 value and selected compounds with lower toxicity [68]. Finally, we screened out the potential molecular drugs of psoriasis following the above three drug design specifications and proposed three potential molecular drugs which are combined as a multi-molecule drug in Table 4. Overall, we have successfully selected a suitable combination of molecular drugs as a multi-molecule drug for the therapeutic treatment of psoriasis.

## 5. Conclusions

In this study, we first constructed candidate GWGENs via big data mining. Second, we used an AIC system order detection and system identification scheme to obtain the real GWGENs of psoriasis and non-psoriasis by their microarray data of 128 patients. Third, we extracted the core GWGENs of psoriasis and non-psoriasis by employing the PNP method. Afterward, core signaling pathways for psoriasis and health control were identified by KEGG pathways annotation of core GWGENs. After identifying the significant biomarkers for the pathogenesis of psoriasis as drug targets, with the help of the DNN-based DTI model trained by DTI databases, we could predict the candidate molecular drugs of the significant biomarkers. Finally, we selected potential molecular drugs to combine as a multi-molecule drug for the therapeutic treatment of psoriasis according to the drug design specifications of adequate regulatory ability, toxicity, and sensitivity between drugs and drug targets.

In this study, we identified NF-κB, FOXO1, CEBPB, STAT3, and ERK1/2 as the drug targets for the treatment of psoriasis. Then based on the prediction of the DNN-based DTI model and the proposed drug design specifications, Naringin, Butein, and Betulinic acid are selected and combined as the multiple-molecule drug to target multiple biomarkers of the pathogenesis of psoriasis. With more clinical and experimental verification, it is hoped that the multi-molecule drug proposed in this study can benefit patients with psoriasis. Finally, it is appealing to extend the novelty of this systems medicine approach in the search for new candidate molecules in the therapy of other diseases since, with the proposed work, the therapeutic expectations of the drugs thus identified are high.

## Figures and Tables

**Figure 1 ijms-24-10033-f001:**
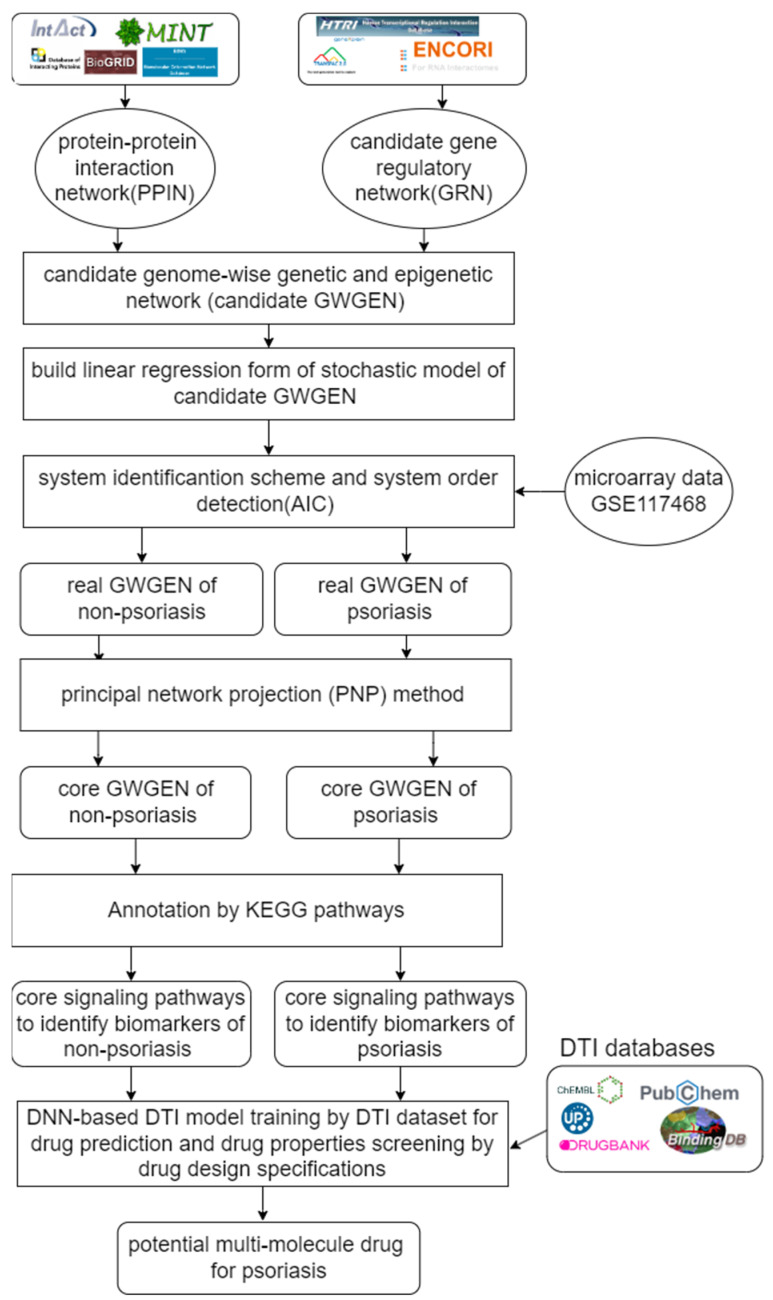
Flowchart of systems biology methods to identify significant biomarkers of pathogenic mechanism as drug targets of psoriasis and the outline of systematic drug discovery design of psoriasis. Candidate GWGEN is composed of a protein–protein interaction network (PPIN) and gene regulatory network (GRN), which are constructed by PPIN and GRN datasets. Real GWGENs were obtained by pruning false positives from candidate GWGENs by microarray data GSE117468 through system order detection and system identification. The core GWGENs were extracted from real GWGENs by the PNP method. Core GWGENs of psoriasis and non-psoriasis are annotated by KEGG pathways to obtain core signaling pathways of psoriasis and non-psoriasis, respectively. The pathogenic biomarkers were identified for the pathogenesis of psoriasis by comparing the core signaling pathways of psoriasis and non-psoriasis. Finally, the potential drugs were discovered by the prediction of the DNN-based DTI model and the screening of drug design specifications. Then, the screened molecular drugs were combined as a multi-molecule drug for the therapeutic treatment of psoriasis.

**Figure 2 ijms-24-10033-f002:**
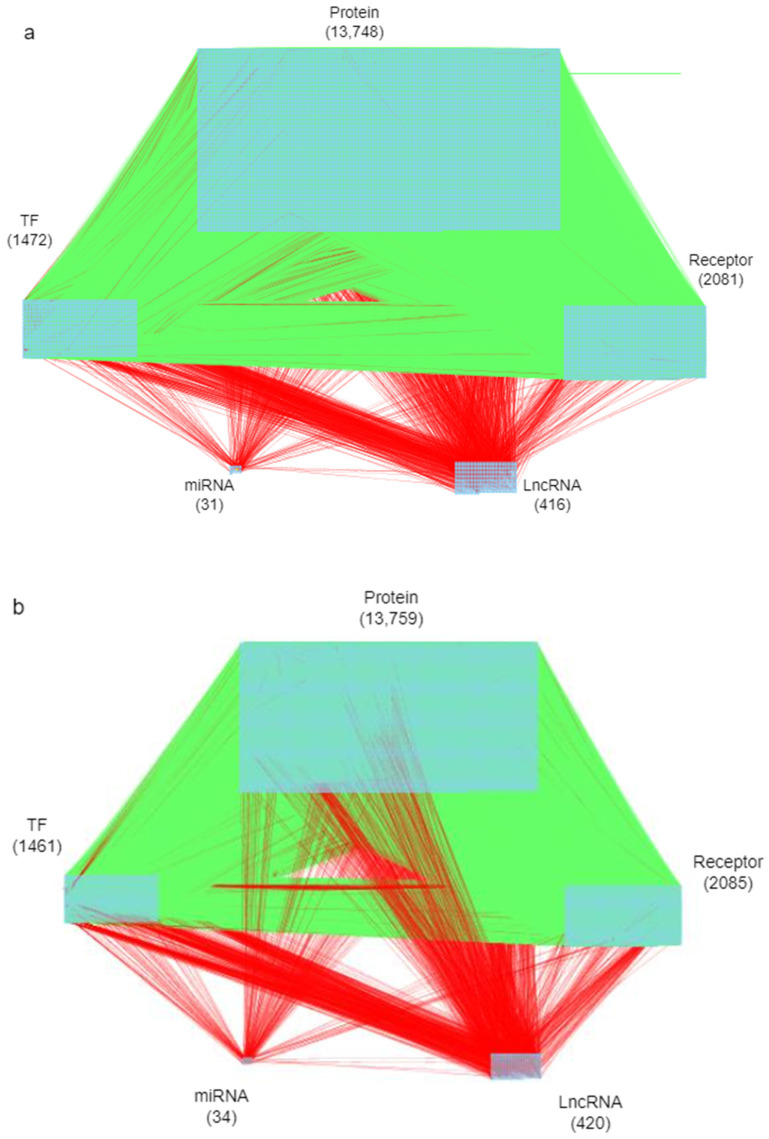
(**a**) The real GWGEN of psoriasis; (**b**) the real GWGEN of non-psoriasis. The green lines represent the PPI, and the red lines represent the gene regulations. Numbers indicate the number of nodes.

**Figure 3 ijms-24-10033-f003:**
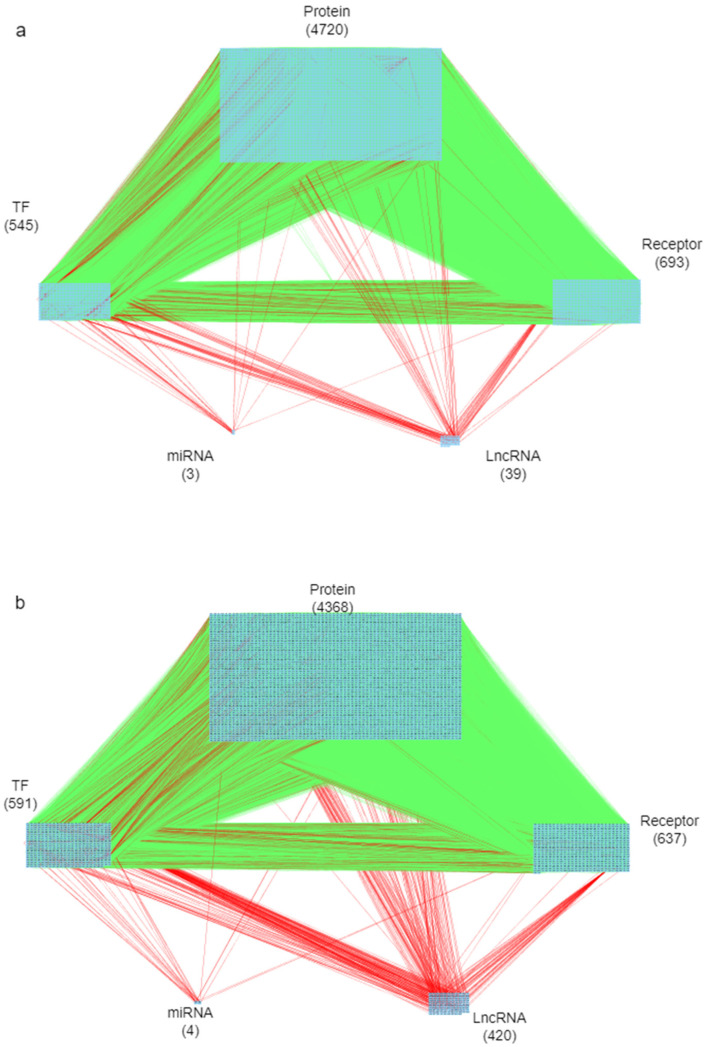
(**a**) The core GWGEN of psoriasis; (**b**) the core GWGEN of non-psoriasis. The green lines represent the PPI, and the red lines represent the gene regulations. Numbers indicate the number of nodes.

**Figure 4 ijms-24-10033-f004:**
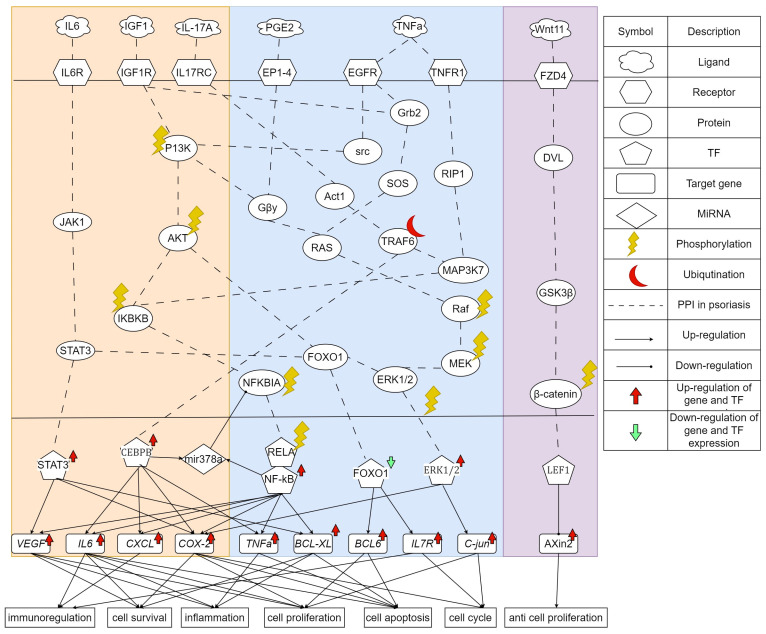
The common and specific core signaling pathways and their downstream cellular dysfunctions between psoriasis and non-psoriasis. The figure shows the core genetic and epigenetic signaling pathways and pathogenic mechanisms of psoriasis. The left block contains the specific core signaling pathways of psoriasis. The middle block contains the overlapping core signaling pathways between psoriasis and non-psoriasis. The right block contains the specific core signaling pathways of non-psoriasis. The gene symbols in red or green font denote the selected significant biomarkers of the pathogenesis of psoriasis as drug targets of psoriasis.

**Figure 5 ijms-24-10033-f005:**
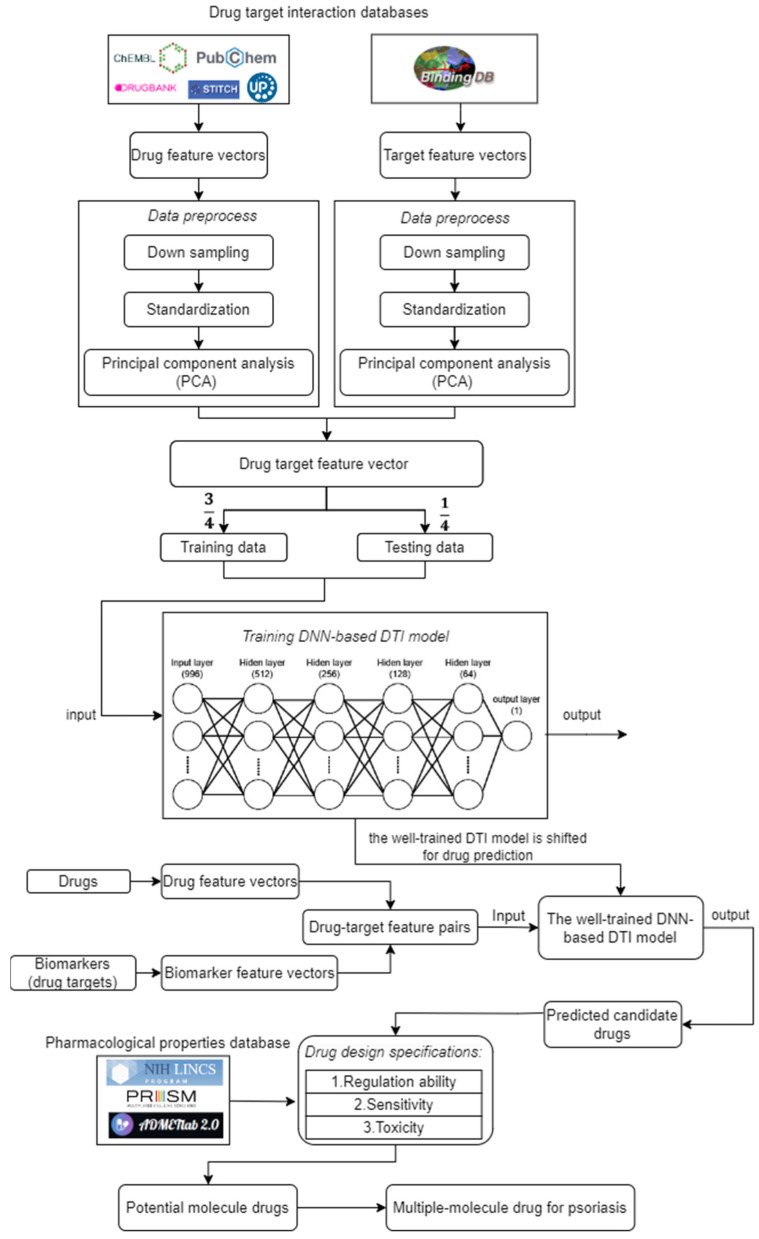
The flowchart of design and discovery of a multi-molecule drug for therapeutic treatment of psoriasis. Drug-target interaction data were obtained from drug-target interaction databases. Then, the drug and target feature vectors were pre-processed, including downsampling, standardization, and PCA, respectively. After data preprocessing, drug target feature vectors were divided into training data and testing data for training the DNN-based DTI model. The well-trained DNN-based DTI model was used to predict candidate drugs for these biomarkers (drug targets). The potential molecule drugs were selected from predicted candidate drugs according to the drug design specifications and combined as multi-molecule drugs for the therapeutic treatment of psoriasis.

**Figure 6 ijms-24-10033-f006:**
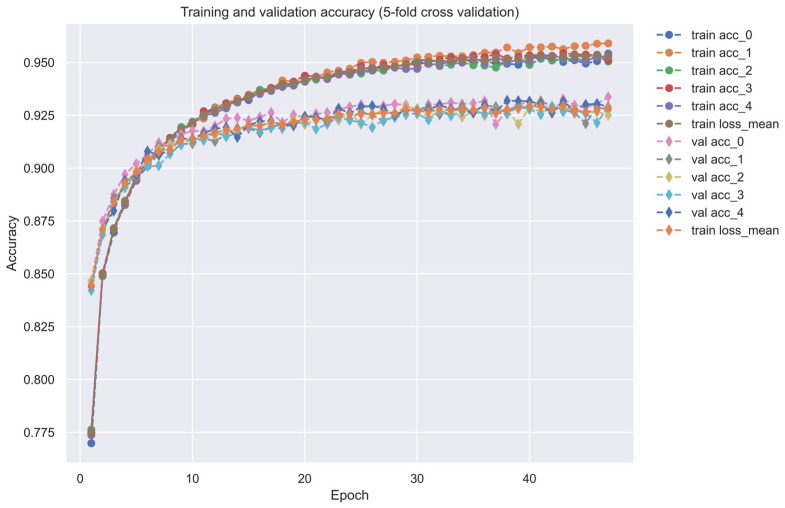
The training and validation accuracy (five-fold cross-validation). ”-o-” line in different colors denotes the training accuracy, “-◊-” line in different colors denotes the validation accuracy.

**Figure 7 ijms-24-10033-f007:**
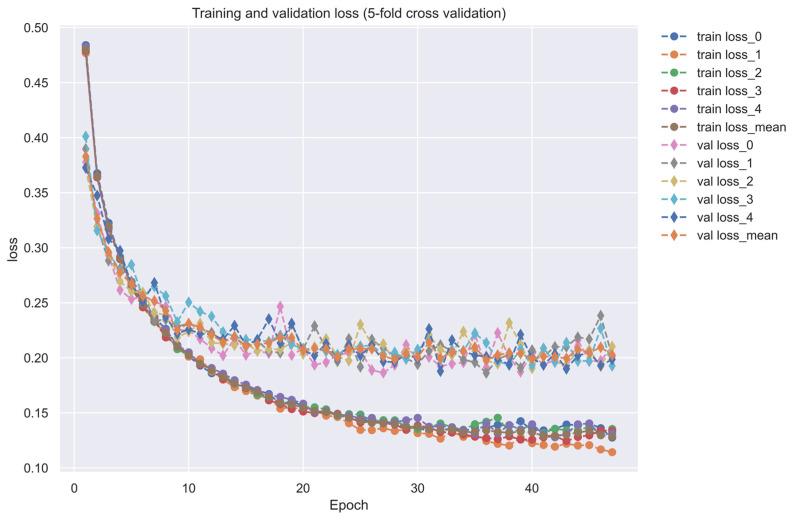
The training and validation loss (five-fold cross-validation).”-o-” line in different colors denotes the training loss, “-◊-” line in different colors denotes the validation loss.

**Figure 8 ijms-24-10033-f008:**
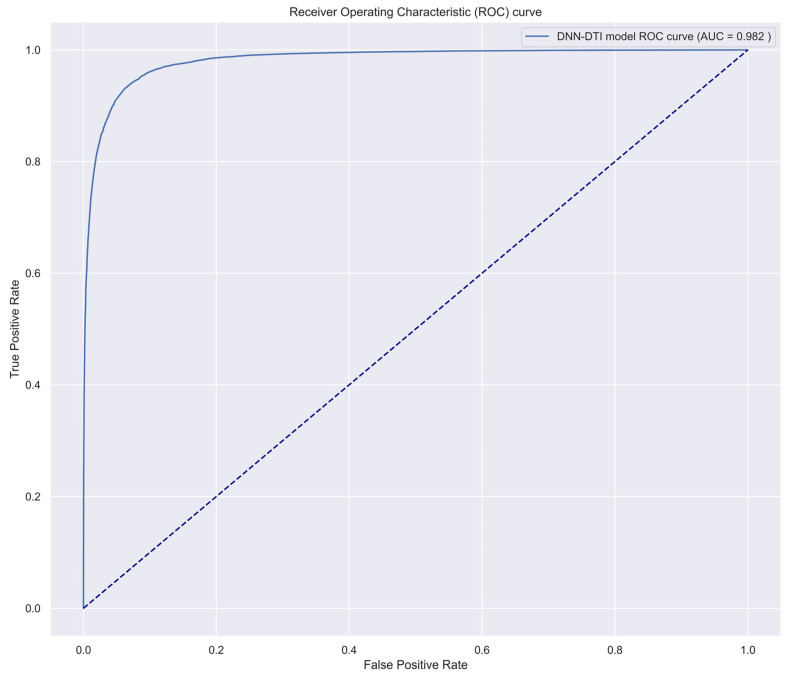
The prediction performance of the trained DNN-based DTI model has an AUC score of 0.982 for its receiver operating characteristic (ROC) curve.

**Table 1 ijms-24-10033-t001:** The statistics of the nodes and edges in real GWGENs of psoriasis and non-psoriasis.

Nodes	Candidate GWGENs of Psoriasis	Real GWGENs of Psoriasis	Real GWGENs of Non-Psoriasis
Receptor	2092	2081	2085
TF	1481	1472	1461
Protein	13,781	13,748	13,759
miRNAs	42	31	34
LncRNAs	436	416	420
Total	17,832	17,748	17,759
**Edges**	**Candidate GWGEN**	**Real GWGEN** **of psoriasis**	**Real GWGEN** **of non-psoriasis**
PPIs	3,872,294	1,711,359	1,712,333
TF-Receptor	14,159	5513	5517
TF-TF	11,808	4289	4439
TF-Protein	79,424	32,426	32,384
TF-miRNA	120	43	52
TF-lncRNA	583	359	356
miRNA-Receptor	694	7	8
miRNA-TF	579	4	9
miRNA-Protein	3969	63	73
miRNA-miRNA	3	1	1
miRNA-lncRNA	22	2	2
lncRNA-Receptor	719	84	74
lncRNA-TF	610	62	77
lncRNA-Protein	4259	852	821
lncRNA-miRNA	2	0	0
lncRNA-lncRNA	15	4	4
Total edges	3,989,260	1,755,068	1,756,150

**Table 2 ijms-24-10033-t002:** KEGG pathways enrichment analysis of core GWGEN of psoriasis.

Pathway	Gene Number	*p*-Value
Pathways in cancer	251	1.2 × 10^−12^
Cellular senescence	92	3.6 × 10^−11^
FoxO signaling pathway	78	7.1 × 10^−10^
Focal adhesion	107	3 × 10^−9^
Cell cycle	74	7.6 × 10^−9^

**Table 3 ijms-24-10033-t003:** KEGG pathways enrichment analysis of core GWGEN of non-psoriasis.

Pathway	Gene Number	*p*-Value
FoxO signaling pathway	76	3.1 × 10^−10^
Colorectal cancer	52	7.8 × 10^−8^
Hippo signaling pathway	82	9.2 × 10^−8^
Cellular senescence	80	3.6 × 10^−7^
Endometrial cancer	38	3.6 × 10^−7^

**Table 4 ijms-24-10033-t004:** The selected potential molecular drugs based on their regulatory ability, toxicity, sensitivity, and their binding of drug targets for the therapeutic treatment of psoriasis.

	Target	FOXO1	NF-kB	STAT3	CEBPB	ERK1/2	Toxicity (LC50)	Sensitivity (PRISM)
Drug								
**Betulinic acid**			**●**		**●**	**●**	**6.712**	**−0.103441**
**Butein**			**●**	**●**			**5.776**	**−0.054450**
**naringin**		**●**	**●**			**●**	**6.921**	**0.132488**
**Chemical structure of multiple molecules drug**								
**Betulinic acid**			**Butein**			**naringin**		
			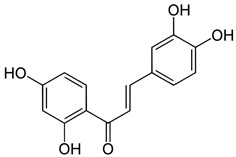			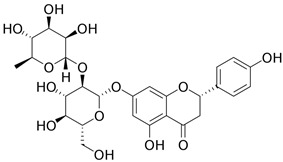		

**Table 5 ijms-24-10033-t005:** Prediction performance of the DNN-based DTI model with five-fold cross-validation (early stopping at epoch 70).

	Validation Loss	Validation Accuracy	Testing Loss	Testing Accuracy
1	0.210332	0.933680	0.216158	0.979626
2	0.201320	0.927631	0.187810	0.983853
3	0.210199	0.925092	0.199039	0.982904
4	0.192665	0.928918	0.214739	0.982421
5	0.198525	0.929007	0.189671	0.982708
average	0.202608	0.928866	0.201483	0.982302
Standard deviation	0.006848	0.002792	0.012029	0.001422

**Table 6 ijms-24-10033-t006:** The pharmacological information of candidate molecular drugs predicted by the DNN-based DTI model in Figure 5 for biomarkers of psoriasis. (+) means overexpression; (−) means low expression in psoriasis.

STAT3 (+)			
Drug	Regulation Ability (L1000)	Sensitivity (PRISM)	Toxicity (LC50, mol/kg)
CCT-018159	−0.68150	0.068573	5.187
Mofezolac	−0.34939	0.027005	5.684
Eliprodil	−2.14303	−0.00262	5.23
Butein	−1.46898	−0.05445	5.776
LY-303511	−0.57210	−0.44921	5.075
**CEBPB (+)**			
**Drug**	**Regulation ability** **(L1000)**	**Sensitivity** **(PRISM)**	**Tox** **icity** **(LC50, mol/kg)**
Limonin	−3.138645411	−0.10563	6.726
Mofezolac	−0.427841723	0.027005	5.684
Betulinic-acid	−0.934850574	−0.10344	6.712
Tangeritin	−0.260343373	−0.00037	5.488
SB-415286	−0.291000009	0.011518	6.282
**ERK1/2 (+)**			
**Drug**	**Regulation ability** **(L1000)**	**Sensitivity** **(PRISM)**	**Toxicity** **(LC50, mol/kg)**
Naringin	−0.12728	0.132488	6.921
Mofezolac	−0.4780038	0.027005	5.684
Betulinic-acid	−0.65759182	−0.10344	6.712
Quercetin	−0.297982544	−0.03326	5.222
BIBU-1361	−0.908186227	0.155512	6.201
**NF-kB (+)**			
**Drug**	**Regulation ability** **(L1000)**	**Sensitivity** **(PRISM)**	**Toxicity** **(LC50, mol/kg)**
Naringin	−0.12728	0.132488	6.921
Mofezolac	−1.00891	0.027005	5.684
Butein	−1.46898	−0.05445	5.776
Ro-04-5595	−0.2619	−0.00387	5.443
Flavoxate	−0.75315	0.107694	5.861
**FOXO1 (−)**			
**Drug**	**Regulation ability** **(L1000)**	**Sensitivity** **(PRISM)**	**Toxicity** **(LC50, mol/kg)**
Naringin	0.579085	0.132488	6.921
Mofezolac	0.748792	0.027005	5.684
Betulinic-acid	0.756684	−0.10344	6.712
Penfluridol	0.892637	0.144082	6.202
Dexamethasone	0.763925	0.122088	5.128

## Data Availability

The raw gene count datasets of human genes are integrated from GSE117468 (https://www.ncbi.nlm.nih.gov/geo/query/acc.cgi?acc=GSE117468, accessed on 15 November 2022). The drug regulation ability data are from Phase I L1000 Level 5 datasets (https://www.ncbi.nlm.nih.gov/geo/query/acc.cgi?acc=GSE92742, accessed on 15 November 2022). The drug sensitivity datasets are from DepMapPRISM primary screen datasets (https://depmap.org/repurposing/, accessed on 15 November 2022). The code can be obtained from https://github.com/ZhanUPing/muti-molecular-drug-design, accessed on 15 November 2022.

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
