# Peer review of "Drug Target Identification and Drug Repurposing in Psoriasis through Systems Biology Approach, DNN-Based DTI Model and Genome-Wide Microarray Data"

_ijms, 2023, doi:10.3390/ijms241210033_

Round 1

Reviewer 1 Report

The paper is very interesting and insightful. The authors analyzed the main pathways involved in the pathogenesis of psoriasis, their interactions and, through data-mining work, constructed a candidate genome-wide genetic and epigenentic network (GWGEN). After that, they extracted the major genes involved, based on the pathogenetic pathways, and then, through a deep neural network (DNN)-based drug-target interaction (DTI) model, selected novel molecular candidates for the treatment of psoriasis.

The work is fully explained, are very detailed mathematical steps adopted by the system used.

The list and interactions of pathogenic pathways are also very useful.

The figures are not always simple, but this is due to the complexity of the topic.

English is clear and well written.

Conclusions on the whole are fine, but I would suggest extending the novelty of this approach in the search for new candidate molecules in therapy, since with such work the therapeutic expectations of the drugs thus identified are high.

Reviewer 2 Report

Comments to the Authors of Manuscript Number ijms-2451079

with the title

Drug target identification and drug repurposing in psoriasis through systems biology approach, DNN-based DTI model and genome-wide microarray data

This is an excellent and valuable article that presents a complex analysis of some mechanisms involved in psoriasis pathology and the drug target identification through systems biology approach. However, corrections are still necessary.

1.     Lines 31-33: please describe the psoriasis lesions more accurately.

2.     Lines 69-71: the steroids are not used in psoriasis systemically, only topically (as dermocorticoids). Please remote or redo these phrases.

3.     Lines 76-78: I don’t understand what is the disease you made reference to in this phrase. Please explain or redo the phrase.  

4.     Lines 80-81: nowadays we have medication that can be used orally in psoriasis, please add. 

5.     Line 142: chapter 2 needs to be material and methods, not the results. The results are after the discussion. Please redo.

- - at figure 1 please correct protien with protein 
